# Pathwise gradient variance reduction with control variates in variational inference

## Abstract

Stochastic gradient descent is a workhorse in modern deep learning. The gradient of interest is almost always the gradient of an expectation, which is unavailable in closed form. The pathwise and score-function gradient estimators represent the most common approaches to estimating the gradient of an expectation. When it is applicable, the pathwise gradient estimator is often preferred over the score-function gradient estimator because it has substantially lower variance. Indeed, the latter is almost always applied with some variance reduction techniques. However, a series of works suggest, in the context of variational inference, that pathwise gradient estimators may also benefit from variance reduction. In this work, we review existing control-variates-based variance reduction methods for pathwise gradient estimators to determine their effectiveness. Work in this vein generally rely on approximations of the integrand which necessitates the functional form of the variational family be simple. In light of this limitation, we also propose applying zero-variance control variates on pathwise gradient estimators, as the control variates have the advantage that requires little assumption on the variational distribution, other than being able to sample from it.

## 1 Introduction

Stochastic optimisation, in particular stochastic gradient descent, is ubiquitous in modern machine learning. In many applications, deployment of stochastic gradient descent necessitates estimating the gradient of an expectation,

$$g(\lambda) = \nabla_\lambda \mathbb{E}_{q(z;\lambda)}[r(z;\lambda)], \tag{1}$$

where $r$ is some real-valued function and $\lambda \in \mathbb{R}^{\dim_\lambda}$ is the parameter over which we optimise. Note that since the expectation is taken with respect to a distribution $q$ that is parameterised by $\lambda$, we cannot simply push the gradient operator through the expectation. The two most common approaches to estimating $g(\lambda)$ are the **pathwise gradient estimator** and the **score-function gradient estimator**. We will introduce them in the context of variational inference (VI).

Given an observed dataset $x$ that is governed by a data generating process which depends on latent variables $z \in \mathbb{R}^{\dim_z}$ and a prior $p(z)$ of the latent variables, the posterior distribution is given by

$$p(z|x) \propto p(x|z)p(z),$$

which is only known up to a normalising constant. VI seeks to approximate the posterior with a simple, tractable distribution in the variational family $\mathcal{Q} = \{q(z;\lambda) : \lambda \in \mathbb{R}^{\dim_\lambda}\}$. This is often done by finding the $\lambda$ that minimises the Kullback-Leibler divergence between the variational distribution $q$ and the posterior (in that order):

$$\lambda^* = \arg\min_\lambda \mathbb{E}_{q(z;\lambda)}[\log q(z;\lambda) - \log p(z|x)]. \tag{2}$$

In practice, this is done by maximising the so-called evidence lower bound (ELBO),

$$\lambda^* = \arg\max_\lambda \text{ELBO}(\lambda), \tag{3}$$

where

$$\text{ELBO}(\lambda) = \mathbb{E}_{q(z;\lambda)}[\log p(z, x) - \log q(z; \lambda)].$$

It is easy to see that minimising the KL divergence in (2) is equivalent to maximising the ELBO in (3). Notably, computation of the latter avoids the intractable normalising constant in the posterior $p(z|x)$.

The closed-form solution for $\lambda^*$ is generally unavailable. Stochastic VI (Hoffman et al., 2013), which optimises with minibatch stochastic gradient descent, became a game changer and opened up new applications for VI. Stochastic VI requires gradient computation of the *minibatch ELBO*,

$$\text{mELBO}(\lambda) = \mathbb{E}_{q(z;\lambda)}\left[r(z; \lambda)\right],$$

where

$$r(z; \lambda) = \frac{N}{B} \sum_{i \in \text{batch}} \left[\log p(x_i|z)\right] + \log \frac{p(z)}{q(z; \lambda)}. \tag{4}$$

Here $N$ and $B$ are data and batch size respectively, and $x_i$ denotes the $i^{th}$ element of the dataset.

## 2    Gradient estimators for VI

There are two main types of gradient estimators in the VI literature for computing the gradient of mELBO: 1) the pathwise gradient, also known as the reparametrization trick; and 2) the score-function estimator, also known as REINFORCE. The latter is more broadly applicable than the former but comes at the cost of larger variance. Indeed, the score-function estimator is almost always used in conjunction with control variates to reduce its variance (see, for example Ranganath et al., 2014; Ji et al., 2021). It is far less common to see variance reduction employed for the pathwise gradient estimator but a series of recent works suggest potential benefits from doing so (Miller et al., 2017; Geffner & Domke, 2020). In this work, we propose a new variance reduction technique for the pathwise gradient estimator in the context of VI. But first in this section we review the pathwise graident estmator.

The pathwise gradient estimator is only readily applicable for *reparametrizable* distributions $q(z; \lambda)$, i.e. distributions where $z$ can be equivalently generated from $z = T(\epsilon; \lambda)$ where $\epsilon \in \mathbb{R}^{\dim_z} \sim q_0(\epsilon)$ and $q_0$ is referred as the *base distribution* which is independent from $\lambda$. Take $z \sim \mathcal{N}(\mu, \sigma^2 I)$ as an example, the corresponding transformation is $T(\epsilon; \lambda) = \mu + \sigma\epsilon$ where $\epsilon$ is standard Gaussian and $\lambda = (\mu, \sigma)$.

When $q$ is reparametrizable, the gradient operator can be pushed inside the expectation and we get that the gradient of mELBO is given by

$$\begin{aligned} g(\lambda) &\coloneqq \nabla_\lambda \text{mELBO}(\lambda) \\ &= \mathbb{E}_{q_0(\epsilon)}\varphi(\epsilon; \lambda), \end{aligned} \tag{5}$$

where we define

$$\varphi(\epsilon; \lambda) = \nabla_\lambda \left[r(T(\epsilon; \lambda); \lambda)\right]. \tag{6}$$

The pathwise gradient estimator is then simply a Monte Carlo estimator of (5) using a set of samples $\{\epsilon_{[l]}\}_{l=1}^L$ from $q_0$:

$$\hat{g}(\epsilon_{[1]}, \dots, \epsilon_{[L]}; \lambda) \coloneqq \frac{1}{L} \sum_{l=1}^L \varphi(\epsilon_{[l]}; \lambda). \tag{7}$$

We will refer to $L$ as the number of gradient samples.

The variance of the gradient estimator is thought to play an important factor in the convergence properties of stochastic gradient descent. Henceforth, any expectations or variances without a subscript refer to $q_0$. Define the variance of the pathwise gradient estimator to be

$$\mathbb{V}[\hat{g}] = \mathbb{E}\|\hat{g}\|^2 - \|\mathbb{E}\hat{g}\|^2 = \frac{1}{L}(\mathbb{E}\|\varphi\|^2 - \|\mathbb{E}\varphi\|^2) = \frac{1}{L}\mathbb{V}[\varphi].$$

To reduce the variance of (7), we may add a *control variate* to the pathwise gradient estimator

$$\frac{1}{L}\sum_{l=1}^{L}\left[\varphi(\epsilon_{[l]};\lambda)+c(\epsilon_{[l]})\right],\tag{8}$$

where the control variance $c(\cdot)\in\mathbb{R}^{\dim_\lambda}$ is a random variable with zero expectation, i.e. $\mathbb{E}[\varphi+c]=\mathbb{E}\varphi$. Let $\mathrm{Tr}(\mathbb{C}[\varphi,c])$ be the trace of the covariance matrix $\mathbb{C}[\varphi,c]=\mathbb{E}[(\varphi-\mathbb{E}\varphi)(c-\mathbb{E}c)^\top]$. A good control variate $c$ has a strong, negative correlation with $\varphi$, since

$$\begin{aligned}
\mathbb{V}[\tfrac{1}{L}\textstyle\sum_{l=1}^{L}\varphi(\epsilon_{[l]};\lambda)+c(\epsilon_{[l]})] &= \tfrac{1}{L}\mathbb{V}[\varphi+c] \\
&= \tfrac{1}{L}(\mathbb{V}[\varphi]+\mathbb{V}[c]+2\,\mathrm{Tr}(\mathbb{C}[\varphi,c])) \\
&= \mathbb{V}[\hat{g}]+\tfrac{1}{L}(\mathbb{V}[c]+2\,\mathrm{Tr}(\mathbb{C}[\varphi,c]))
\end{aligned}\tag{9}$$

Therefore, as long as $\mathrm{Tr}(\mathbb{C}[\varphi,c])<0$ and $|\mathrm{Tr}(\mathbb{C}[\varphi,c])|<\tfrac{1}{2}\mathbb{V}[c]$, the control-variate-adjusted gradient estimator (8) will have a smaller variance than (7).

Finally, a control variate may also be formed as a linear combinations of control variates. Let $C:\mathbb{R}^{\dim_z}\to\mathbb{R}^{\dim_\lambda\times J}$ be a matrix with $J$ control variates in the columns. This leads to the control-variate-adjusted gradient estimator

$$\hat{h}(\epsilon_{[1]},\dots,\epsilon_{[L]};\lambda):=\frac{1}{L}\sum_{l=1}^{L}\left[\varphi(\epsilon_{[l]};\lambda)+C(\epsilon_{[l]})\beta\right],\tag{10}$$

which remains a valid control variate due to the linearity of expectation operators, i.e. $\mathbb{E}[C\beta]=0$. Here $\beta\in\mathbb{R}^J$ is a vector of coefficients corresponding to each control variate. This construction allows us to combine multiple weak control variates into a strong one by adjusting $\beta$.

For obvious reasons, applying control variates is only worthwhile if the computation of control variates is cheaper than increasing the number of samples in (7). From (9), we can see that, for example, the estimator variance can be reduced by half either by doubling $L$ or halving $\mathbb{V}[\varphi+c]$. This presents a unique challenge when applying control variates in the low $L$ regime (such as the gradient estimator of VI where $L$ is often very low), since the cost of applying control variates will more likely outweigh the cost of increasing $L$ to achieve the same variance reduction. The control variates that were developed in the Markov Chain Monte Carlo (MCMC) community cannot be easily applied in our work because 1) they require large $L$, but $L$ can be as small as one in stochastic VI, and 2) MCMC variance reduction is usually done at the very end, while variance reduction is required for each gradient update in stochastic VI.

In this work, we review existing control-variates-adjusted pathwise gradient estimators in the context of VI. We are primarily interested in whether employing control variates can achieve faster convergence with respect to wall-clock time. We are also motivated by the gap in the VI literature on gradient variance reduction when $q$ is reparametrizable but the mean and covariance of $q$ are not available in closed form. A good example of such $q$ is normalizing flow, where $z$ is the result of pushing forward a base distribution $q_0$ through an invertible transformation $T(\cdot;\lambda)$ that is parameterised by $\lambda$, i.e. $z=T(\epsilon;\lambda)$ where $\epsilon\sim q_0$. Such transformation $T$ can be arbitrarily complex and usually involves neural networks. To this end, we introduce a control-variate-adjusted gradient estimator based on zero-variance control variates (Assaraf & Caffarel, 1999; Mira et al., 2013; Oates et al., 2017) which does not have this limitation.

This paper is structured as follows: in Section 3, we provide a review of the latest advancements in variance reduction techniques for pathwise gradient estimators in the context of VI. This is followed by a discussion on methods for selecting $\beta$ and $C$ of (10) in Sections 4 and 5, respectively. The novel method based on zero-variance control variates is introduced in Section 5.2. Finally, we present the experimental results in Section 6.

## 3 Related work

Variance reduction for the pathwise gradient estimator in VI has been explored in Miller et al. (2017) and Geffner & Domke (2020). These works focused on designing a single control variate (i.e. $C$ only has a column)

with the form of $C = \mathbb{E}\tilde{\varphi} - \tilde{\varphi}$, where $\tilde{\varphi}(\epsilon; \lambda)$ is an approximation of $\varphi(\epsilon; \lambda)$. The expectation $\mathbb{E}\tilde{\varphi}$ is designed to be theoretically tractable, but this usually implies that there is some restriction imposed on $T$ (and therefore $q$) to make this expectation easy to compute.

Miller et al. (2017) proposed $\tilde{\varphi}$ that is based on the first-order Taylor expansion of $\nabla_z \log p(z, x)$. However, this Taylor expansion requires the expensive computation of the Hessian $\nabla_z^2 \log p(z, x)$. Geffner & Domke (2020) improved upon their work and proposed using a quadratic function to approximate $\log p(z, x)$. Their method only requires the first-order gradient $\nabla_z \log p(z, x)$, and their $\mathbb{E}\tilde{\varphi}$ is available in closed-form as a function of the mean and covariance of $q$. A direct modification of their method is to estimate $\mathbb{E}\tilde{\varphi}$ empirically when the mean and covariance of $q$ are unavailable; see Section 5.1 for more details. Both Miller et al. (2017) and Geffner & Domke (2020) only considered Gaussian $q$ in their work, although the latter can be applied to a larger class of variational families where the mean and covariance of $q$ is known.

The proposed estimator based on zero-variance control variates is similar in spirit to another work from the same group in Geffner & Domke (2020). Like Geffner & Domke (2018), we propose combining weak control variates into a stronger one, but our work differs in the construction of the individual control variates and the optimisation criterion for $\beta$.

## 4 Selecting $\beta$ for the control-variate-adjusted pathwise gradient estimator

The utility of control variates depends on the choice of $\beta$ and $C$ in (10). In this section, we will discuss various strategies to pick an appropriate $\beta$ given a family of control covariates $C$.

### 4.1 A unique set of $\beta$ for each dimension of $\lambda$

The formulation in (10) suggests that the same set of $\beta$ is used across the dimensions of $\varphi$. This can be too restrictive for $C$ that are weakly-correlated to $\varphi$. In such instance, having a unique set of $\beta$ coefficients for each dimension of $\varphi$ can be beneficial, as it allows the coefficients to be selected on a per-dimension basis. In fact, this can be easily done by turning $C$ into a $\dim_\lambda \times (J\dim_\lambda)$-dimensional, block diagonal matrix

$$\begin{bmatrix} C_{1,:} & \dots & 0 \\ \vdots & \ddots & \vdots \\ 0 & \dots & C_{\dim_\lambda,:} \end{bmatrix},$$

where $C_{i,:}$ is the $i^{th}$ row of the original $C$. In other words, we expand the number of control variates to $J\dim_\lambda$, and each control variate will only reduce the variance of a single dimension of $\varphi$.

### 4.2 Optimisation criteria for $\beta$

The $\beta$ is usually chosen to minimise the variance of $\hat{h}$. In practice, this variance is usually replaced by an empirical approximation due to the lack of its closed-form expression. There are three approximations in the literature, the first of which is a direct approximation of the variance with samples $\{\epsilon_{[l]}\}_{l=1}^L$,

$$\mathbb{V}[\varphi + C\beta] \approx \tfrac{1}{L(L-1)} \sum_{l > l'} \|\varphi(\epsilon_{[l]}) + C(\epsilon_{[l]})\beta - \varphi(\epsilon_{[l']}) - C(\epsilon_{[l']})\beta\|^2, \tag{11}$$

as seen in Belomestny et al. (2018). The second approximation is based on the definition of variance

$$\mathbb{V}[\varphi + C\beta] = \mathbb{E}\|\varphi - \mathbb{E}[\varphi + C\beta] + C\beta\|^2 \approx \min_{\alpha \in \mathbb{R}^{\dim_\lambda}} \tfrac{1}{L} \sum_{l=1}^L \|\varphi(\epsilon_{[l]}) + \alpha + C(\epsilon_{[l]})\beta\|^2, \tag{12}$$

where $\alpha \in \mathbb{R}^{\dim_\lambda}$ is an intercept term in place of the unknown $\mathbb{E}[\varphi + C\beta]$. The second approximation in (12) is generally cheaper to compute than the first approximation in (11) as it only requires $O(L)$ operations rather than $O(L^2)$; see Si et al. (2022) for details.

Minimising (12) with respect to $\beta$ is essentially a least squares problem with a well-known closed-formed solution

$$\begin{bmatrix} \alpha^* \\ \beta^* \end{bmatrix} = \arg\min_{\alpha,\beta} \sum_{l=1}^{L} \left\| \varphi(\epsilon_{[l]}) + \begin{bmatrix} \mathbb{I}_{\dim_\lambda} & C(\epsilon_{[l]}) \end{bmatrix} \begin{bmatrix} \alpha \\ \beta \end{bmatrix} \right\|^2 \tag{13}$$

$$= -(\boldsymbol{X}^\top \boldsymbol{X})^{-1} \boldsymbol{X}^\top \boldsymbol{\varphi}, \tag{14}$$

where $\mathbb{I}_{\dim_\lambda}$ is an identity matrix of size $\dim_\lambda$, and $\boldsymbol{\varphi}$ and the design matrix $\boldsymbol{X}$ are given by

$$\boldsymbol{\varphi} = \begin{bmatrix} \varphi(\epsilon_{[1]}) \\ \vdots \\ \varphi(\epsilon_{[L]}) \end{bmatrix}, \quad \boldsymbol{X} = \begin{bmatrix} \mathbb{I}_{\dim_\lambda} & C(\epsilon_{[1]}) \\ \vdots & \vdots \\ \mathbb{I}_{\dim_\lambda} & C(\epsilon_{[L]}) \end{bmatrix}.$$

However, the matrix inversion in (14) can be problematic to compute, especially when $\boldsymbol{X}^T \boldsymbol{X}$ is rank-deficient. In such instance, we can include a penalty in (13) and solve for the penalised least squares solution (see, for example, South et al., 2022), or use an iterative optimisation algorithm to minimise (13), as suggested in Si et al. (2022).

Finally the third approach relies on the assumption that $\mathbb{E}[C\beta] = 0$ and is based on the observation that

$$\mathbb{V}[\varphi + C\beta] = \mathbb{E}\|\varphi + C\beta\|^2 - \|\mathbb{E}\varphi\|^2.$$

This suggests that the variance can be equivalently minimised by minimising simply the expected squared norm component, $\mathbb{E}\|\varphi + C\beta\|^2$. Geffner & Domke (2018) shows that the minimiser $\beta^* = \arg\min_\beta \mathbb{E}\|\varphi + C\beta\|^2$ is given by

$$\beta^* = -\mathbb{E}[C^\top C]^{-1} \mathbb{E}[C^\top \varphi], \tag{15}$$

and suggests replacing $\mathbb{E}[C^\top C]$ and $\mathbb{E}[C^\top \varphi]$ with their empirical estimates. This approach, however, requires inverting a costly inversion of size $J$ matrix.

### 4.3 Potential bias in the gradient estimator

The unbiasedness of the control-variate-adjusted Monte Carlo estimator (10) relies on the assumption that the $\beta$ are independent of $C$, since $\mathbb{E}[C(\epsilon)\beta(\epsilon)] \neq 0$ in general. This necessitates that $\beta$ and $C$ should be estimated with independent sets of $\epsilon$ samples. However, in practice, the $\beta$ is estimated with the same set of $\epsilon$ in $C$ to save computational time at the cost of introducing bias in the gradient estimates.

## 5 Control variates

Having reviewed methods to select $\beta$ given a family of control variates $C$, we now turn our attention to constructing $C$. We will first propose a simple modification of Geffner & Domke (2020) that will enable it to work for variational distributions $q$ with unknown mean and covariance. Subsequently, we will introduce zero-variance control variates, which can be constructed without the knowledge of $q$ or $T$.

### 5.1 Quadratic approximation control variates

In this section we review the quadratic-approximation control variates proposed in Geffner & Domke (2020). An important distinction at the outset is their assumption that the entropy term in mELBO,

$$-\mathbb{E}_{q(z)} \log q(z; \lambda),$$

is *known.* As such the focus of Geffner & Domke (2020) is to reduce the variance of

$$\mathbb{E}\nabla_\lambda f(T(\epsilon; \lambda)),$$

where

$$f(z) = \frac{N}{B} \sum_{i \in \text{batch}} [\log p(x_i|z)] + \log p(z). \tag{16}$$

Geffner & Domke (2020) proposed control variates of the form

$$C(\epsilon) = \mathbb{E}[\nabla_\lambda \tilde{f}(T(\epsilon; \lambda))] - \nabla_\lambda \tilde{f}(T(\epsilon; \lambda)), \tag{17}$$

where

$$\tilde{f}(z; v) = b_v^\top (z - z_0) + \frac{1}{2}(z - z_0)^\top B_v (z - z_0)$$

is a quadratic approximation of (16). Here, $v = \{B_v, b_v\}$ are the parameters of the quadratic equation that are chosen to minimise the $L^2$ difference between $\nabla f(z)$ and $\nabla \tilde{f}(z)$. We will drop $v$ in $\tilde{f}$ for the sake of brevity. The location parameter $z_0$ is set to $\mathbb{E}T(\epsilon; \lambda)$. This quadratic approximation of $f$ can also be viewed as a linear approximation on $\nabla f$.

The first term in (17) has a closed-form expression that depends on the mean and covariance of $q(z; \lambda)$, making the expectation cheap to evaluate when they are readily available. However, this is not the case when $T(\epsilon; \lambda)$ is arbitrarily complex, e.g. normalizing flow. A direct workaround of this limitation is to replace $\mathbb{E}\nabla_\lambda \tilde{f}(T(\epsilon; \lambda))$ with its empirical estimate based on a sample of $\epsilon$'s. Note that $\tilde{f}$ requires $z_0 = \mathbb{E}T(\epsilon; \lambda)$, which we estimate using another independent set of $\epsilon$'s. See Algorithm 1 for a summary of the procedure.

As (16) is a part of the Monte Carlo estimator (10), it could be tempting to estimate $\mathbb{E}\nabla_\lambda \tilde{f}(T(\epsilon; \lambda))$ with an average of the $\nabla_\lambda \tilde{f}(T(\epsilon; \lambda))$ evaluations that have been computed in (10). This is to be avoided as it will result in the two terms in (17) cancelling each other out.

As (17) is designed to be strongly correlated with $\varphi$ when $\tilde{f}$ is reasonably close to $f$, the choice of $\beta$ becomes less significant. Geffner & Domke (2020) opted to minimise the expected squared norm with a scalar $\beta$ (note that $C$ is a column vector in this case), the solution of which is given in (15). In their work, the expectations $\mathbb{E}[C^\top \varphi]$ and $\mathbb{E}[C^T C]$ are replaced with their empirical estimates computed from $C$ and $\varphi$ in (10), instead of fresh evaluations. However, as discussed in Section 4.2, the resulting gradient estimate is biased due to the dependency of between $\beta$ and $C$.

While this bias is not mentioned explicitly in Geffner & Domke (2020), we conjecture that they overcame the issue by estimating the expectations with $C$ and $\varphi$ computed from previous iterations, as specified in Algorithm 1. Therefore, their $\beta$ is independent from $C$ in the current iteration. This will avoid introducing bias to the gradient estimate at the cost of having sub-optimal $\beta$. They also claimed that their estimates of $\mathbb{E}[C^\top \varphi]$ and $\mathbb{E}[C^T C]$ (and by extension, $\beta$) do not differ much across iterations. Moreover, their $\beta$ is largely acting as an auxiliary 'switch' of the control variate when $\tilde{f}$ is a poor approximation of $f$, rather than the primary mechanism to reduce the estimator variance, since the $\beta$ will be almost 0 when $\tilde{f}$ is not approximating well (i.e. $\mathbb{C}[\varphi, C] \approx 0$). Their $C$ only kicks in when it is sufficiently correlated to $\varphi$.

Lastly, let us return to the discussion of the entropy term at the beginning of this section. Our setup is more general than Geffner & Domke (2020) as we does not assume the entropy term $-\mathbb{E}_{q(z)} \log q(z; \lambda)$ to necessarily have a closed-form expression, i.e. our $r(z, \lambda)$ includes $-\log q(z; \lambda)$. Although it was claimed in Geffner & Domke (2020) that their quadratic approximation control variate can also be similarly designed for $r(z, \lambda)$ in (4) rather than $f(z, \lambda)$ in (16), we found the implementation difficult because the updating step of $v$ requires the gradient $\nabla_z \log q(z; \lambda)$, and in turn $\frac{\partial \lambda}{\partial z}$, which is challenging to compute.

## 5.2 Zero-variance control variates

The control variates in Geffner & Domke (2020) require one to know the mean and covariance of $q(z; \lambda)$. To avoid this requirement, we propose the use of gradient-based control variates (Assaraf & Caffarel, 1999; Mira et al., 2013; Oates et al., 2017). These control variates are generated by applying a so-called Stein operator $\mathcal{L}$ to a class of user-specified functions $P(z)$. Typically the Stein operator uses $\nabla_z \log q(z)$, the gradients of the log probability density function for the distribution over which the expectation is taken, but it does not require any other information about $\varphi$ or $T$.

---

**Algorithm 1** Quadratic approximation control variates with empirical estimates of $\mathbb{E}\tilde{f}$

---

**Require:** Learning rates $\gamma^{(\lambda)}$, $\gamma^{(v)}$.
  Initialise $\lambda$, $v$ and control variate weight $\beta = 0$.
  **for** $k = 0, 1, 2, \cdots$ **do**
    Sample $\epsilon_{[1]}, \ldots \epsilon_{[L]} \sim q_0$ to compute $\varphi(\epsilon_{[l]}; \lambda_k)$
    Generate an independent set of 100 $\epsilon$ samples to estimate $z_0 = \mathbb{E}T(\epsilon; \lambda)$
    Generate an independent set of 100 $\epsilon$ samples to estimate $\mathbb{E}\nabla_\lambda \tilde{f}(T(\epsilon; \lambda); v_k)$
    Compute $h = \frac{1}{L}\sum_{l=1}^{L}\left[\varphi(\epsilon_{[l]}; \lambda_k) + C(\epsilon_{[l]})\beta\right]$          See (17)
    Take an ascent step $\lambda_{k+1} \leftarrow \lambda_k + \gamma^{(\lambda)}h$
    Estimate $\mathbb{E}[C^\top C]$ and $\mathbb{E}[C^\top \varphi]$ with $\epsilon_{[1]}, \ldots \epsilon_{[L]}$, and update $\beta$ with (15).
    Take a descent step $v_{k+1} \leftarrow v_k - \gamma^{(v)}\frac{1}{2L}\sum_{l=1}^{L}\nabla_v\|\nabla_z f(T(\epsilon_{[l]}; \lambda_k)) - \nabla_z\tilde{f}(T(\epsilon_{[l]}; \lambda_k); v_k)\|^2$
  **end for**

---

We will focus on the form of gradient-based control variates known as zero-variance control variates (ZVCV, Assaraf & Caffarel, 1999; Mira et al., 2013). ZVCV uses the second order Langevin Stein operator and a polynomial $P(z) = \sum_{j=1}^{J}\beta_j P_j(z)$, where $P_j(z)$ is the $j$th monomial in the polynomial and $J$ is the number of monomials. The control variates are

$$\{\mathcal{L}P_j(z)\}_{j=1}^{J} = \{\Delta_z P_j(z) + \nabla_z P_j(z) \cdot \nabla_z \log q(z)\}_{j=1}^{J},$$

where $\Delta_z$ is the Laplacian operator and $q(z)$ is the probability density function for the distribution over which the expectation is taken. A sufficient condition for these control variates to have zero expectation is that the tails of $q$ decay faster than polynomially (Appendix B of Oates et al., 2016), which is satisfied by Gaussian $q$ for example.

In this paper, we only consider first-order polynomials so there are $J = \dim_z$ control variates of the form

$$\left\{\frac{\partial}{\partial z_j}\log q(z)\right\}_{j=1}^{\dim_z}.$$

Here, $z_j$ refers to the $j^{th}$ dimension of $z$. We do not find second-order polynomials to have any advantage over first-order polynomials; see Appendix F for a discussion. For pathwise gradient estimators using a standard Gaussian as the base distribution, these control variates simplify further to

$$\left\{\frac{\partial}{\partial \epsilon_j}\log q_0(\epsilon)\right\}_{j=1}^{\dim_z} = \{-\epsilon_j\}_{j=1}^{\dim_z}.$$

We are also using the same set of control variates across different dimensions of $\varphi$, but assigning each dimension with a unique set of $\beta$. That is, the matrix of control variates $C$ is a block-diagonal matrix of size $\dim_\lambda \times \dim_\lambda\dim_z$

$$C(\epsilon) = \begin{bmatrix} -\epsilon^\top & \ldots & 0 \\ \vdots & \ddots & \vdots \\ 0 & \ldots & -\epsilon^\top \end{bmatrix}. \tag{18}$$

This is in contrast to Geffner & Domke (2020) where the values in $C$ is different across dimensions, but their $\beta$ is shared. The simplicity of ZVCV comes with the drawback that it is often not as correlated as the integrand. This makes the choice of $\beta$ crucial.

### 5.2.1 Exact least squares

As discussed in Section 4.2, the optimal $\beta$ can be selected by solving (13), the solution of which is given in (14). We can further exploit the block-diagonal structure of (18) and decompose (13) into a series of linear

regression that corresponds to each dimension of $\lambda$. In other words, we can solve $\beta$ for each dimension of $\lambda$ individually

$$\begin{bmatrix} \alpha_i^* \\ \beta_i^* \end{bmatrix} = -(\boldsymbol{X}^\top \boldsymbol{X})^{-1} \boldsymbol{X}^\top \boldsymbol{\varphi}_i, \quad i = 1, \ldots, \dim_\lambda, \tag{19}$$

where $i$ denotes the dimension of $\lambda$ which $\alpha_i^*$, $\beta_i^*$ and $\boldsymbol{\varphi}_i$ correspond to, e.g. $\boldsymbol{\varphi}_i = [\varphi_i(\epsilon_{[1]}), \ldots, \varphi_i(\epsilon_{[L]})]^\top$ is a subset of $\boldsymbol{\varphi}$ in (14) that corresponds to the $i^{th}$ dimension of $\lambda$, and

$$\boldsymbol{X} = \begin{bmatrix} 1 & -\epsilon_{[1]}^\top \\ \vdots & \vdots \\ 1 & -\epsilon_{[L]}^\top \end{bmatrix}.$$

Note that the inversion of $\boldsymbol{X}^\top \boldsymbol{X}$ only needs to be performed once and can be used across different $\varphi_i$, thereby scaling well to models with high-dimensional $\lambda$. The control variate for $\varphi_i$ can then be computed as $\varphi_i(\epsilon) = -\epsilon^\top \beta_i^*$. We also propose using $\epsilon_{[1]}, \ldots, \epsilon_{[L]}$ that have already been generated in (10) to compute $\beta^*$, as the resulting control variate tends to have a lower mean squared error.

### 5.2.2 Least squares with gradient descent

There are several challenges we face in applying ZVCV. For example, $\boldsymbol{X}^T \boldsymbol{X}$ in (19) must be invertible. This is often not the case for models with large $\dim_z$, as we usually keep $L$ low and thus the column space of $\boldsymbol{X}$ is rank-deficient. This can be solved by adding a penalty term in (13) (i.e. shrinking $\beta$ towards 0), and empirical evidence suggests that is more effective than using a subset of these control variates to achieve better performance (Geffner & Domke, 2018; South et al., 2022). However, penalised least squares remains prohibitively expensive to solve as it still involves inverting a matrix of size $\dim_z$.

Instead, we propose mimicking penalised least squares by minimising (13) with respect to $\alpha$ and $\beta$ with gradient descent. This is done by

1. Initialise $\alpha$ at $-\frac{1}{L} \sum_{l=1}^{L} \varphi(\epsilon_{[l]})$ and $\beta$ at the zero vector. Set $\gamma^{(\alpha,\beta)}$ to a low value;

2. Take a descent step $(\alpha_{m+1}, \beta_{m+1}) \leftarrow (\alpha_m, \beta_m) - \gamma^{(\alpha,\beta)} \frac{1}{L} \sum_{l=1}^{L} \nabla_{\alpha,\beta} \|\varphi(\epsilon_{[l]}) + \alpha_m + C(\epsilon_{[l]})\beta_m\|^2$;

3. Repeat Step 2 for a few times.

See Algorithm 2 for a complete description. The combination of learning rate and number of iterations is analogous to the penalty in penalised least squares: a lower number of iterations and learning rate $\gamma^{(\alpha,\beta)}$ will result in a near-zero $\beta$ that results from a stronger penalty (more shrinkage of $\beta$ towards 0). This procedure is also similar in spirit to Si et al. (2022).

## 6 Experiments

In these experiments, we assess the efficacy of various control variate strategies.

**Model and datasets** We perform VI on the following model-dataset pairs: logistic regression on the *a1a* dataset, a hierarchical Poisson model on the *frisk* dataset, and Bayesian neural network (BNN) on the *redwine* dataset. For the BNN model, we consider a full-batch gradient estimator trained on a subset of 100 data points of the *redwine* dataset following the experimental setup in Geffner & Domke (2020) and Miller et al. (2017). We also consider a mini-batch estimator of size 32 but trained on the full *redwine* dataset, see Appendix A for more details. With the exception of mini-batch BNN, these models appeared in either Geffner & Domke (2020) or Miller et al. (2017).

**Variational families** Three classes of variational families are considered:

- **Mean-field Gaussian** The covariance of the Gaussian distribution $\mathcal{N}(\mu, \Sigma)$ is parameterised by log-scale parameters, i.e. $\Sigma = \text{diag} \left( \exp(2 \log \sigma_1, \ldots, 2 \log \sigma_{\dim_z}) \right)$.

---

**Algorithm 2** ZVCV-GD

---

**Require:** Learning rates $\gamma^{(\lambda)}$, $\gamma^{(\alpha,\beta)}$.

  Initialise $\lambda$

  **for** $k = 0, 1, 2, \cdots$ **do**

    Sample $\epsilon_{[1]}, \ldots \epsilon_{[L]} \sim q_0$

    Compute $\varphi(\epsilon_{[l]}; \lambda_k), \forall l = 1, \ldots, L$                            See (5)

    Initialise $\alpha_0 = -\frac{1}{L} \sum_{l=1}^{L} \varphi(\epsilon_{[l]}; \lambda_k)$ and $\beta_0$ at the zero vector

    **for** $m = 0, 1, 2, \cdots$ **do**

      Take a descent step $(\alpha_{m+1}, \beta_{m+1}) \leftarrow (\alpha_m, \beta_m) - \gamma^{(\alpha,\beta)} \frac{1}{L} \sum_{l=1}^{L} \nabla_{\alpha,\beta} \| \varphi(\epsilon_{[l]}; \lambda_k) + \alpha_m + C(\epsilon_{[l]})\beta_m \|^2$

    **end for**

    Set $\beta^*$ as the final value of $\beta$ from the previous inner loop

    Compute $h = \frac{1}{L} \sum_{l=1}^{L} \varphi(\epsilon_{[l]}; \lambda_k) + C(\epsilon_{[l]})\beta^*$                See (18)

    Take an ascent step $\lambda_{k+1} \leftarrow \lambda_k + \gamma^{(\lambda)} h$.

  **end for**

---

- **Rank-5 Gaussian** The covariance of the Gaussian distribution $\mathcal{N}(\mu, \Sigma)$ is parameterised by a factor $F \in \mathbb{R}^{\dim_z \times 5}$ and diagonal components, i.e. $\Sigma = FF^\top + \mathrm{diag}\left(\exp(2 \log \sigma_1, \ldots, 2 \log \sigma_{\dim_z})\right)$.

- **Real NVP** We use a real NVP normalizing flow (Dinh et al., 2017) with two coupling layers and compose the layers in alternate pattern. The flow has a standard multivariate Gaussian as its base distribution. The scale and translation networks have the same architecture of $8 \times 16 \times 16$ hidden units with ReLU activations, followed by a fully connected layer. There is an additional tanh activation at the tail of the scale network to prevent the exponential term from blowing up.

We only present the results for mean-field Gaussian and real NVP in the main section. The results for rank-5 Gaussian are included in Appendix C, as they are largely similar to those obtained for mean-field Gaussian.

**Optimiser and learning rate**   We use an Adam optimiser and set its learning rate $\gamma^{(\lambda)} = 0.01$, except for the BNN models with real NVP where we set $\gamma^{(\lambda)} = 0.001$. These learning rates have been selected as the most best options, in terms of convergence time to a respectable ELBO, from the set of $\{0.1, 0.01, 0.001, 0.0001\}$.

**Control variates**   The gradient estimator is equipped with the following control variate strategies:

- **NoCV** The vanilla gradient estimator without any control variates.

- **ZVCV-GD** A ZVCV with $\beta$ minimising least squares with an inner gradient descent, as described in Algorithm 2 and Section 5.2.2. We set the learning rate $\gamma^{(\alpha,\beta)} = 0.001$ and iterated the inner gradient descent 4 times for each outer Adam step. These hyperparameter choices may not always yield the maximum variance reduction in every situation, but they represent a good compromise with computation time. Additionally, we have discovered that prolonging the inner gradient descent iterations does not necessarily lead to better variance reduction. For a more comprehensive discussion, please refer to Appendix F.

- **QuadCV** This is the original algorithm presented in Geffner & Domke (2020) when $q$ is Gaussian (i.e. the mean and covariance of $q$ are readily available). When $q$ is real NVP, we use Algorithm 1 and 100 samples to estimate $\mathbb{E} T(\epsilon; \lambda)$ and $\mathbb{E} \nabla_\lambda \tilde{f}(T(\epsilon; \lambda))$. The learning rate $\gamma^{(v)}$ is set to $\gamma^{(\lambda)}$, following the original work.

Note that above we only compare our method in detail with Geffner & Domke (2020) as it is a direct improvement of Miller et al. (2017).

**Initialisations** We repeated the experiment five times, each time using different initialisations of $\lambda$ to assess the convergence performance of each method under varying initial conditions. For the mean-field Gaussian, the $\lambda$ values were randomly sampled from a zero-mean Gaussian distribution with a scale parameter of 0.5. In contrast, for real NVP, we initialised the $\lambda$ values using a Glorot normal initialiser (Glorot & Bengio, 2010). These choices of initialisers were made deliberately to ensure a wide range of initial values, covering both favourable and unfavourable starting points. Consequently, we expect to observe a diverse range of ELBO trajectories.

**Evaluation settings** We report the ELBO and variance of the gradient estimators. The ELBO for evaluation purpose is always computed with the full dataset (even when using mini-batched ELBO for optimisation) and 500 samples from $q$. We also present the variance ratio, $\mathbb{V}[\hat{h}]/\mathbb{V}[\hat{g}]$ where $\hat{g}$ and $\hat{h}$ as defined in (7) and (10) respectively, in every 50 iterations; a ratio less than 1 indicates a reduction in variance relative to the corresponding NoCV with the same number of $L$. The variance of the gradient estimators is computed by repeatedly sampling 100 gradients (say, $\hat{g}_{[1]}, \ldots, \hat{g}_{[100]}$) from the estimator and computed with $\mathbb{V}[\hat{g}] \approx \frac{1}{100} \sum_j \|\hat{g}_{[j]} - (\frac{1}{100} \sum_i \hat{g}_{[i]})\|^2$. See Appendix B for more details on the calculation of the variance ratio.

## 6.1 ELBO against iteration counts

The results in Figure 1 demonstrate that QuadCV generally outperform NoCV, while ZVCV-GD provides only marginal improvement and can even converge to a suboptimal maximum in some cases (e.g. logistic regression, real NVP 2, and $L = 10$). The performance gap between the estimators also decreases as the number of gradient samples $L$ increases, as seen in the bottom rows of Figure 1a and 1b. It should be noted that QuadCVs may perform poorly in the early stages of gradient descent (e.g. logistic regression on mean-field Gaussian and hierarchical Poisson on real NVP) as it takes time to learn the quadratic function $\tilde{f}$. In general, there is also a high degree of variability in ELBO across different runs. This is especially noticeable in Figure 1a due to the substantial impact of $\lambda$ initialisation on optimisation convergence. For a more detailed examination of the individual trajectories with various initialisations, please refer to Appendix E.

The variance ratio of the gradient estimators can help explain the performance gap observed in Figure 1. As shown in Figure 2, QuadCV generally achieves a lower variance than ZVCV-GD, particularly for Gaussian $q$ when $\mathbb{E}\tilde{f}$ can be computed exactly. The estimator with ZVCV-GD and larger $L$ tends to perform better in models with fewer control variates (i.e. low $\dim_z$), as the $\beta$ is less susceptible to overfitting when solving the least squares with the gradient descent algorithm discussed in Section 5.2.2. On the contrary, in models with large $\dim_z$, such as BNNs, ZVCV-GD fails to reduce variance.

A noteworthy characteristic of QuadCV is that variance reduction only becomes prominent after $\tilde{f}$ in (17) has been adequately trained. This typically occurs as the optimisation process nears convergence. With a QuadCV-adjusted gradient estimator, it is possible to push the ELBO at convergence a few nats further, although significant time has to be spent to reach convergence at all. However, this raises an interesting question about the worthiness of such an effort, as a relatively minor improvement in ELBO may not necessarily translate into substantially improved downstream metrics; see Appendix E for a more in-depth discussion.

The comparison between $L = 10$ and $L = 50$ in Figure 1 suggests that variance reduction in the early stages can facilitate quicker convergence in terms of iteration counts (notice the leftward shift in the trajectories for $L = 50$. This observation implies that employing a larger number of gradient samples is an effective strategy to improve the convergence performance of stochastic VI, as long as the computation of additional gradient samples remains cost-effective in the overall optimisation process. It is important to note that increasing $L$ from 10 to 50 immediately reduces the gradient estimator's variance by five-fold (equivalent to a variance ratio of 0.2) from the very first iteration of the optimisation, in contrast to QuadCV. These results suggest that variance reduction is more beneficial during the initial stages of optimisation when the goal is to expedite convergence towards a satisfactory ELBO, rather than aiming to attain the maximum achievable ELBO.

## 6.2 ELBO against wall-clock time

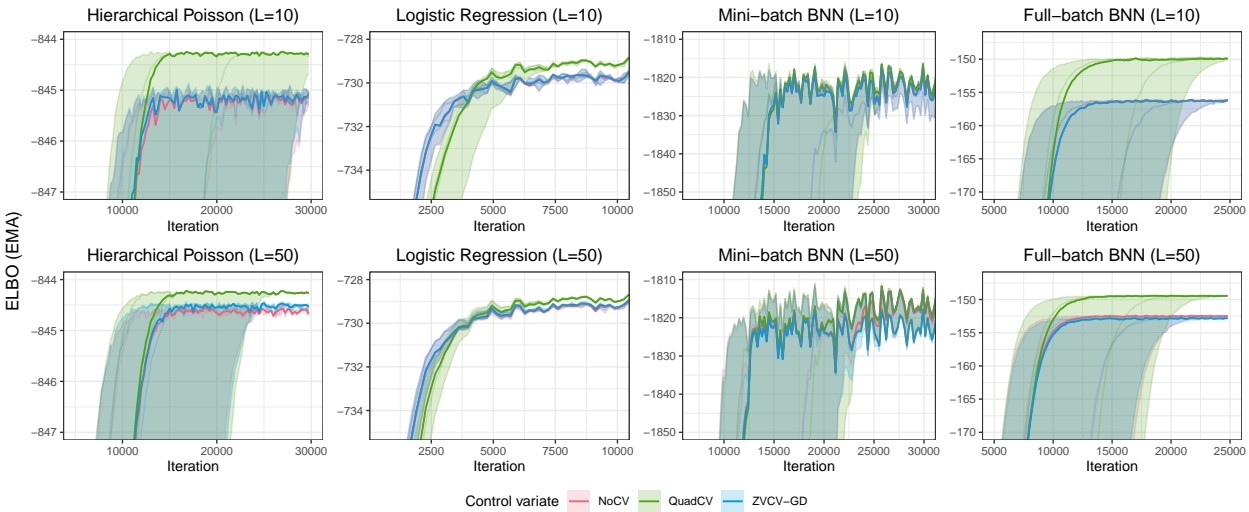

(a) Mean-field Gaussian with 10 (top) and 50 (bottom) gradient samples.

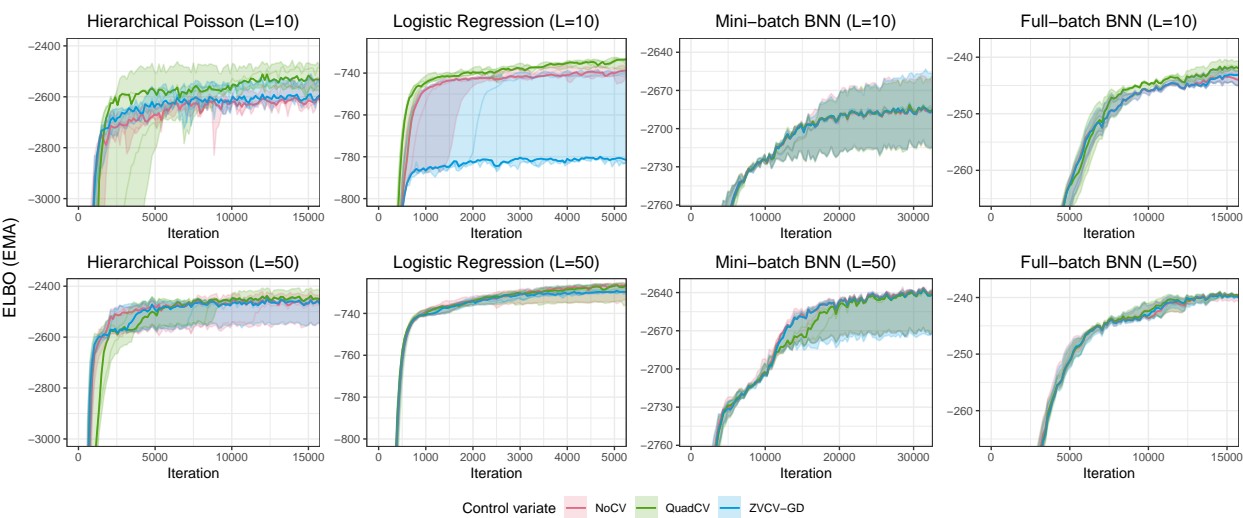

(b) Real NVP with 10 (top) and 50 (bottom) gradient samples.

Figure 1: ELBO is plotted against the number of gradient descent steps for different numbers of gradient samples $L$ and two families of $q$. The bold lines represent the median of ELBO values recorded at the same iteration across five repetitions. The shaded area illustrates the range of ELBO values across five repetitions. The ELBO values are smoothed using an exponential moving average. The trajectories of ZVCV-GD and NoCV are nearly identical in both full-batch and mini-batch BNN when $L = 10$. A higher ELBO indicates better performance. See Figure 6 for plots where the bold lines represent the mean ELBO.

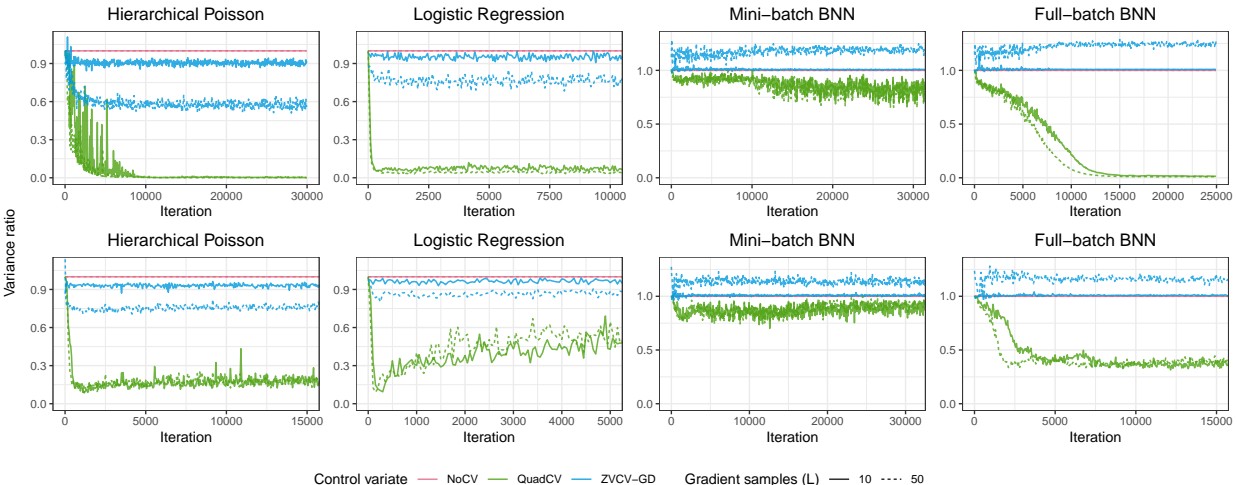

Figure 2: We present the variance ratio $\mathbb{V}[\hat{h}]/\mathbb{V}[\hat{g}]$, where $\hat{g}$ is NoCV and $\hat{h}$ is either ZVCV-GD or QuadCV, at each iteration. We show only the median variance ratios recorded at the same iteration across five repetitions, omitting the individual variance ratios from each repetition to prevent clutter in the plots. The ratios from mean-field Gaussian and real NVP are shown in top and bottom rows respectively. Note that NoCV (in red) is always 1 by definition. We see that ZVCV-GD (in blue) struggles to reduce variance in the BNN models. There is also a significant overlap in QuadCV between $L = 10$ (solid green) and $L = 50$ (dotted green). A lower ratio indicates better performance. See Figure 7 for plots where the bold lines represent the mean variance ratios.

To assess whether the computational expense of calculating control variates or additional gradient samples justifies the potential improvement in ELBO, we measure ELBO against wall-clock time, as illustrated in Figure 3. We timed our VI implementation in JAX and ran on an Nvidia A100 80GB GPU. It is worth noting that recorded times may vary among computing platforms and implementations, given that our code was compiled with XLA (resulting in platform-dependent binaries) and ran without memory constraints.

Our experiments reveal that NoCV generally converges to a respectable ELBO more swiftly. Furthermore, the performance gap between the estimators is even narrower when $L = 50$. An unexpected observation is that increasing $L$ from 10 to 50 incurs negligible computational cost but produce meaningfully faster convergence, as evident when comparing the top and bottom rows of Figure 3a and 3b. It is important to note that the computational cost of extra gradient samples may vary depending on the construction of $\varphi$, and increasing $L$ might not always be a worthwhile strategy for achieving faster convergence (see, for example, the BNNs experiments in Figure 4b of Appendix C).

QuadCV does succeed in increasing the maximum achievable ELBO in certain scenarios, albeit at the expense of longer convergence times. For instance, QuadCV can improve ELBO by approximately 0.7 nats and 6 nats in hierarchical Poisson and full-batch BNN when using a mean-field Gaussian $q$ at $L = 10$. However, this comes at a cost of roughly 50% to 100% more runtime compared to NoCV. Given finite computational resources and the absence of a universal guarantee that a slight ELBO increase will substantially enhance downstream metrics (as discussed in, for example, Yao et al., 2018; 2019; Foong et al., 2020; Masegosa, 2020; Deshpande et al., 2022), it is left to practitioners to determine whether implementing control variates is a worthwhile endeavour.

## 7 Conclusion

In our study of the pathwise gradient estimator in VI, we reviewed the existing state-of-the-art control variates for reducing gradient variance, namely the QuadCV in Geffner & Domke (2020). We identified a gap in the literature of variance reduction of pathwise gradient estimators in stochastic VI resulting from the

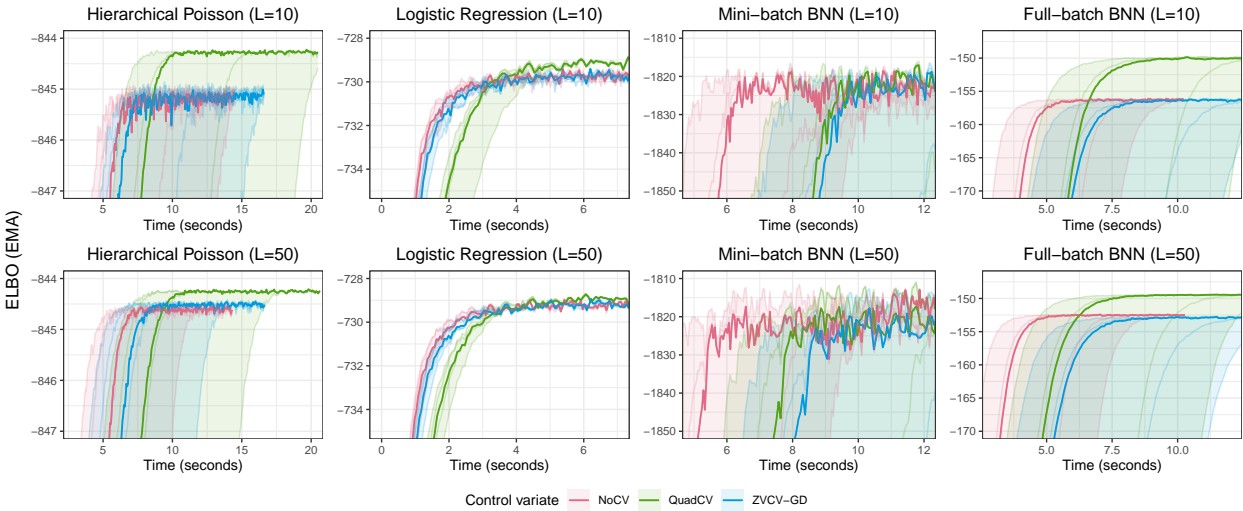

(a) Mean-field Gaussian with 10 (top) and 50 (bottom) gradient samples.

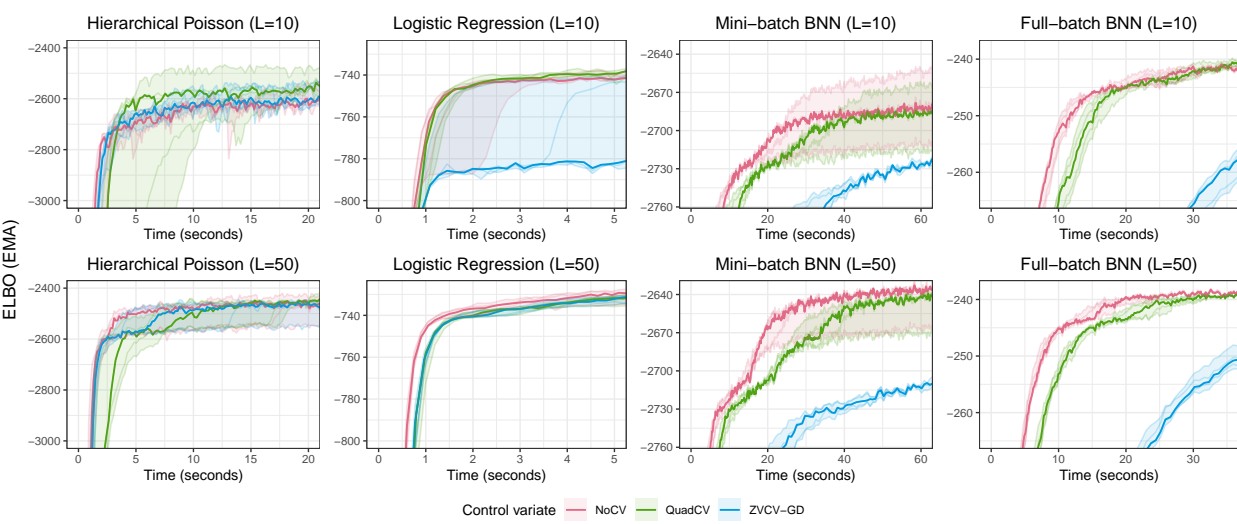

(b) Real NVP with 10 (top) and 50 (bottom) gradient samples.

Figure 3: ELBO is plotted against wall-clock time for different numbers of gradient samples $L$ and two families of $q$. The bold lines represent the median of ELBO values recorded at the same iteration across five repetitions. The shaded area illustrates the range of ELBO values across five repetitions. The ELBO values are smoothed using an exponential moving average. A higher ELBO indicates better performance. See Figure 8 for plots where the bold lines represent the mean ELBO.

setting where the variational distribution has intractable mean and covariance, rendering the state-of-the-art represented by Geffner & Domke (2020) not applicable directly. To address this gap, we proposed using ZVCV, which does not assume specific conditions on the variational distribution. However, our empirical results showed that neither the ZVCV-adjusted nor the QuadCV-adjusted estimator provided substantial improvement against our evaluation criteria that justifies their implementation. Instead, we found that increasing the number of gradient samples is a highly cost-effective method for improving convergence time.

Taking a step back, it is worth discussing the fundamental value in performing variance reduction for pathwise gradient estimators in stochastic VI. For one, it is quite interesting that a dramatic reduction in gradient variance can fail to deliver any discernible effect on the ELBO. This is what was observed in the experiments section — even when the variance ratio was substantially lower than 1, the control variate-adjusted gradient estimator, compared to the vanilla gradient estimator, did not move the needle in a meaningful manner on the ELBO optimisation objective. As such, it can be expected that downstream metrics including log pointwise predictive density or predictive mean squared error will also reveal the general futility of equipping the gradient estimator with a control variate. These findings seem to point to a negative phenomenon for pathwise gradients in stochastic VI — reducing the gradient variance is insufficient to improving downstream performance.

In future work, we hope to explore ZVCV-adjusted gradient estimators in generative models where it can truly shine. Namely, ZVCV is particularly powerful when the distribution of interest is difficult to sample from. One class of distribution models that fit this description are energy-based models (Song & Kingma, 2021).

Relatedly, there is a class of stochastic VI methods known as implicit VI. The variational distribution employed is still required to be reparametrizable but we drop the requirement that the so-called pathwise score, $\nabla_z \log q(z; \lambda)$, be known, e.g. as in normalizing flows. It was shown in Titsias & Ruiz (2019) that the pathwise score may itself be written as an expectation, $\nabla_z \log q(z; \lambda) = E_{q(\epsilon|z;\lambda)} \nabla_z \log q(z|\epsilon; \lambda)$ where $q(\epsilon|z; \lambda)$ is referred to as the reverse conditional. In Titsias & Ruiz (2019), the expectation with respect to the reverse conditional is based on MCMC samples. We could conceivably improve the efficiency by employing ZVCV here.

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

## A    Models and datasets

**Logistic regression with the *a1a* dataset**    We extracted the $a1a$ dataset from the repository hosting Geffner & Domke (2020). We used the full dataset $\{\boldsymbol{x}_i, y_i\}_{i=1}^{1605}$ and 90% of the dataset for training. The response $y_i$ is binary and is modelled as

$$w_0, \boldsymbol{w} \sim \mathcal{N}(0, 10^2)$$

$$p(y_i|\boldsymbol{x}_i, z) = \text{Bernoulli}\left(\frac{1}{1 + \exp(-w_0 - \boldsymbol{w}^T \boldsymbol{x}_i)}\right),$$

where $z = \{w_0, \boldsymbol{w}\}$ and $\dim_z = 120$. The size of training and test sets are 1440 and 165 respectively.

**Hierarchical Poisson regression with the *frisk* dataset**    This example is coming from Gelman et al. (2007). We only used a subset of data (weapon-related crime, precincts with 10%-40% of black proportion), as in Miller et al. (2017) and Geffner & Domke (2020). The response $y_{ep}$ denotes the number of frisk events due to weapons crimes within an ethnicity group $e$ in precinct $p$ over a 15-months period in New York City:

$$\mu \sim \mathcal{N}(0, 10^2)$$
$$\log \sigma_\alpha, \log \sigma_\beta \sim \mathcal{N}(0, 10^2)$$
$$\alpha_e \sim \mathcal{N}(0, \sigma_\alpha^2)$$
$$\beta_p \sim \mathcal{N}(0, \sigma_\beta^2)$$
$$\log \lambda_{ep} = \mu + \alpha_e + \beta_p + \log N_{ep}$$
$$p(y_{ep}|z) = \text{Poisson}(\lambda_{ep}),$$

where $z = \{\alpha_1, \alpha_2, \beta_1, \ldots, \beta_{32}, \mu, \log \sigma_\alpha, \log \sigma_\beta\}$ and $\dim_z = 37$. $N_{ep}$ is the (scaled) total number of arrests of ethnicity group $e$ in precinct $p$ over the same period of time. We do not split out a test set due to its small size (total data size is 96).

**Bayesian neural network with the *redwine* dataset**    We push a vector input $\boldsymbol{x}_i$ through a 50-unit hidden layer and ReLU activation's to predict wine quality. The response $y_i$ is an integer from 1 to 10 (inclusive) measuring the score of red wine. We place an uniform improper prior on the log-variance of the weights and error (see Section 5.7 of Gelman et al., 2013, for a discussion on the prior choice):

$$p(\log \alpha^2) \propto 1, \quad \text{equivalent to } p(\alpha) \propto \alpha^{-1}$$
$$p(\log \tau^2) \propto 1, \quad \text{equivalent to } p(\tau) \propto \tau^{-1}$$
$$w_i \sim \mathcal{N}(0, \alpha^2), \quad i = 1, \ldots, 651$$
$$y_i|\boldsymbol{x}_i, \boldsymbol{w}, \tau \sim \mathcal{N}(\phi(\boldsymbol{x}, \boldsymbol{w}), \tau^2)$$

where $\phi$ is a multi-layer perception. Here, $z = \{\log \alpha^2, \log \tau^2, \boldsymbol{w}\}$ and $\dim_z = 653$. For full-batch gradient descent, we use two mutually exclusive subsets of 100 data point as train and test sets, as in Miller et al. (2017) and Geffner & Domke (2020). For mini-batch gradient descent, we use 90% of the full dataset for training and the rest for testing (size of train and test sets are 1431 and 168 respectively).

## B    Computation of variance ratio

The variance ratio $\mathbb{V}[\hat{h}]/\mathbb{V}[\hat{g}]$ was computed with the following step:

1. Collect 100 samples of $\hat{g}$ resulting in $\{\hat{g}_{[j]}\}_{j=1}^{100}$;

2. For each $\hat{g}_{[j]}$, compute its corresponding control-variate-adjusted gradient estimate $\hat{h}$ (10) to collect $\{\hat{h}_{[j]}\}_{j=1}^{100}$;

3. Estimate $\mathbb{V}[\hat{g}] \approx \frac{1}{100} \sum_j \|\hat{g}_{[j]} - (\frac{1}{100} \sum_i \hat{g}_{[i]})\|^2$. Repeat the same step for $\mathbb{V}[\hat{h}]$;

4. Calculate the ratio $\mathbb{V}[\hat{h}]/\mathbb{V}[\hat{g}]$.

This ratio is designed to evaluate the effectiveness of control variates in reducing variance relative to a corresponding gradient estimator without control variates. Therefore, in our work, the ratio is always computed with a pair of $\hat{g}$ and $\hat{h}$ with the same number of samples.

## C  Results from rank-5 Gaussian

The insight derived from Figure 4 and 5 below are similar to those obtained from Figure 1, 3 and 2. In most cases, the cost for evaluating control variates outweighs the improvement in ELBO achieved through variance reduction in the gradient estimator. We observe marginal gain in ELBO despite the estimators with control variates taking longer time to converge.

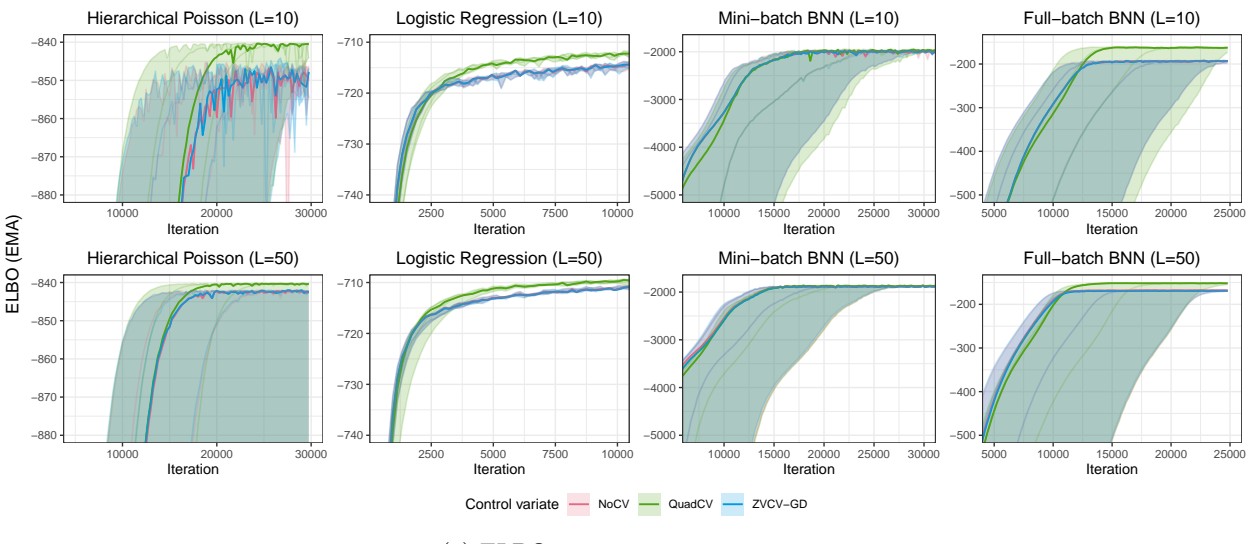

(a) ELBO versus iteration counts.

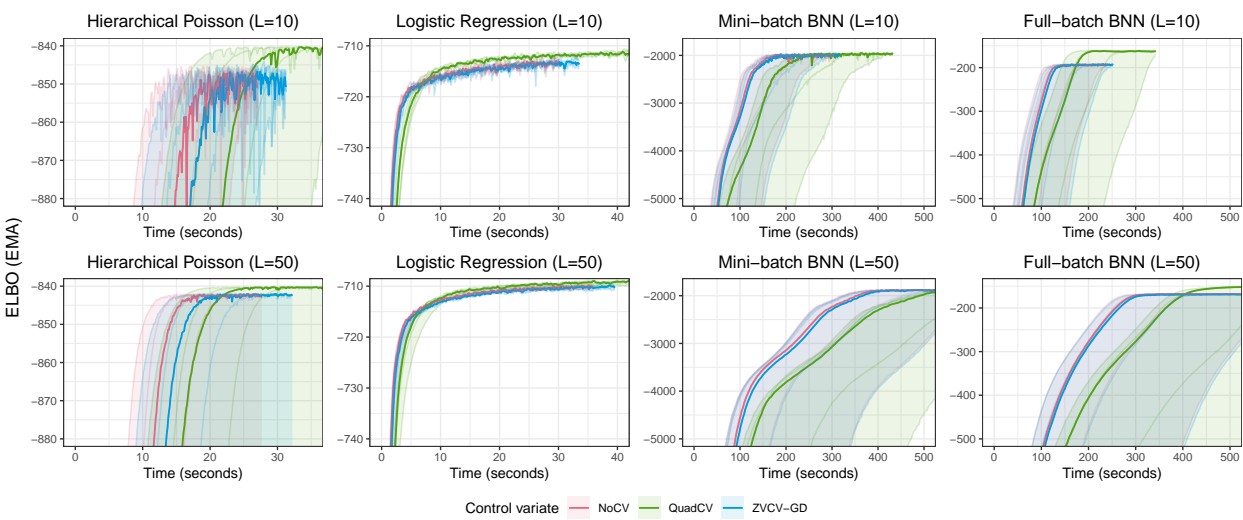

(b) ELBO versus wall-clock time.

Figure 4: ELBO is plotted against gradient descent steps and wall-clock time for varying numbers of gradient samples $L$ using rank-5 Gaussian. The bold lines represent the median of ELBO values recorded at the same iteration across five repetitions. The ELBO values have been smoothed using an exponential moving average. A higher ELBO indicates better performance. See Figure 9 for plots where the bold lines represent the mean ELBO.

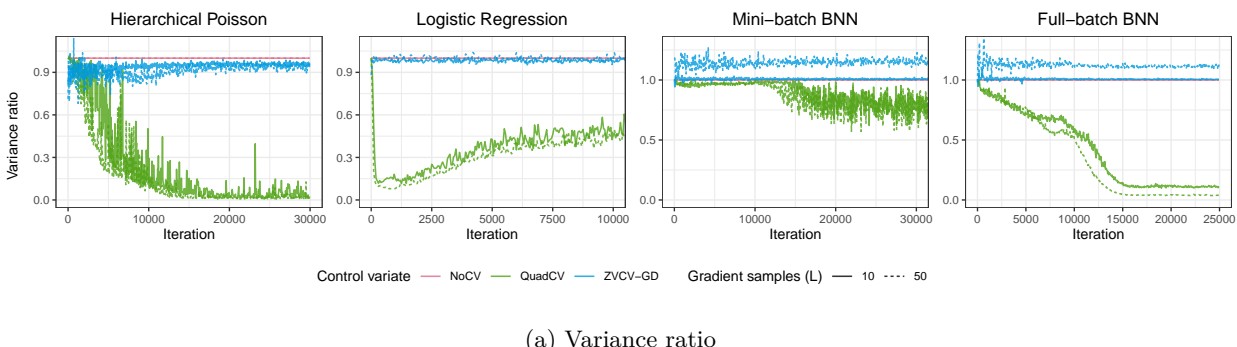

(a) Variance ratio

Figure 5: We present the variance ratio $\mathbb{V}[\hat{h}]/\mathbb{V}[\hat{g}]$ of rank-5 Gaussian, where $\hat{g}$ is NoCV and $\hat{h}$ is either ZVCV-GD or QuadCV, at each iteration. We show only the median variance ratios recorded at the same iteration across five repetitions, omitting the individual variance ratios from each repetition to prevent clutter in the plots. Note that NoCV (in red) is always 1 by definition. We see that ZVCV-GD (in blue) struggles to reduce variance in the BNN models. There is also some overlap between $L = 10$ (solid green) and $L = 50$ (dotted green). A lower ratio indicates better performance. A lower ratio indicates better performance. See Figure 10 for plots where the bold lines represent the mean variance ratios.

# D Mean ELBO trajectories and variance ratio

We have recreated the figures in Section 6 and Appendix C, with the exception that the bold lines now represent the means of ELBO or variance ratios, as opposed to their medians. Using means provides a more transparent depiction of the robustness of each method, although it can be substantially influenced by the repetition that starts farthest from the optimal $\lambda$. Ideally, individual trajectories should be plotted separately (as in Appendix E), but this is not feasible due to space limitations. Nonetheless, the findings of this study are substantiated by interpreting either the mean or median of the evaluation statistics.

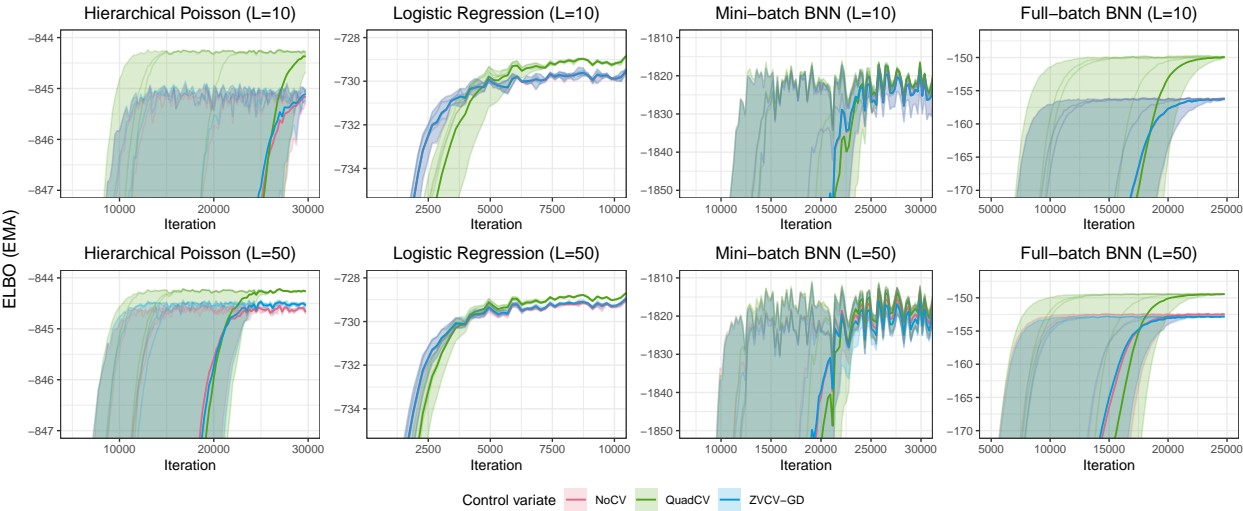

(a) Mean-field Gaussian with 10 (top) and 50 (bottom) gradient samples.

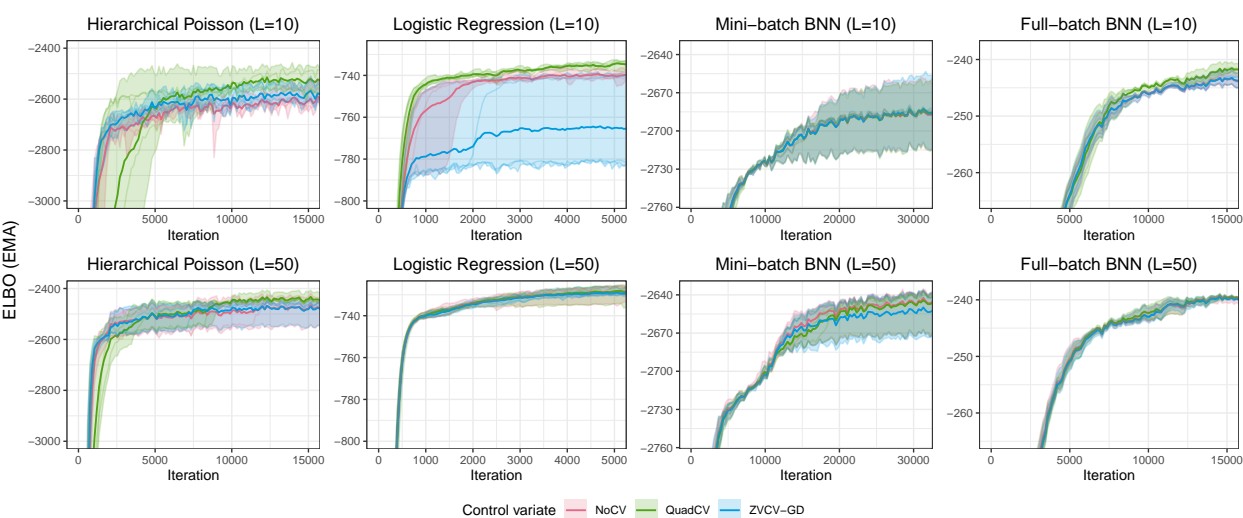

(b) Real NVP with 10 (top) and 50 (bottom) gradient samples.

Figure 6: ELBO is plotted against the number of gradient descent steps for different numbers of gradient samples $L$ and two families of $q$. The bold lines represent the mean of ELBO values recorded at the same iteration across five repetitions. The shaded area illustrates the range of ELBO values across five repetitions. The ELBO values are smoothed using an exponential moving average. The trajectories of ZVCV-GD and NoCV are nearly identical in both full-batch and mini-batch BNN when $L = 10$. A higher ELBO indicates better performance. See Figure 1 for plots where the bold lines represent the median ELBO.

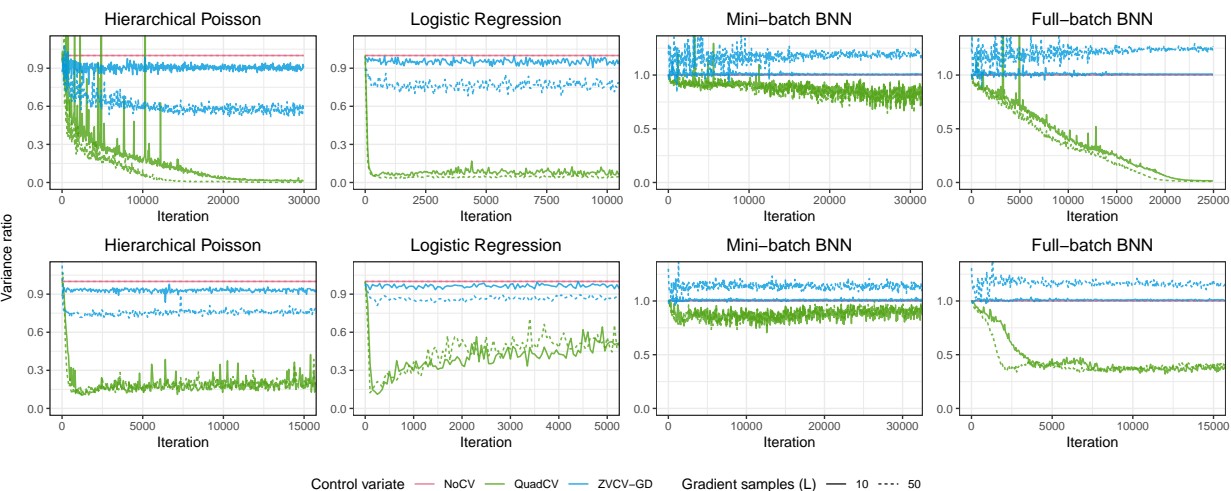

Figure 7: We present the variance ratio $\mathbb{V}[\hat{h}]/\mathbb{V}[\hat{g}]$, where $\hat{g}$ is NoCV and $\hat{h}$ is either ZVCV-GD or QuadCV, at each iteration. We show only the mean variance ratios recorded at the same iteration across five repetitions, omitting the individual variance ratios from each repetition to prevent clutter in the plots. The ratios from mean-field Gaussian and real NVP are shown in top and bottom rows respectively. Note that NoCV (in red) is always 1 by definition. We see that ZVCV-GD (in blue) struggles to reduce variance in the BNN models. There is also a significant overlap in QuadCV between $L = 10$ (solid green) and $L = 50$ (dotted green). A lower ratio indicates better performance. See Figure 2 for plots where the bold lines represent the median variance ratios.

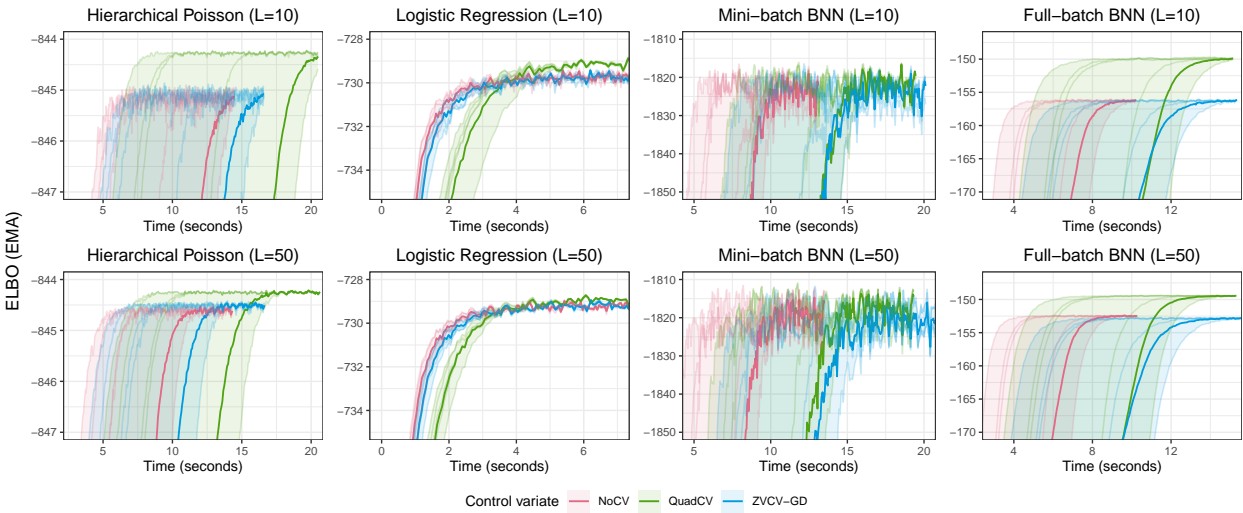

(a) Mean-field Gaussian with 10 (top) and 50 (bottom) gradient samples.

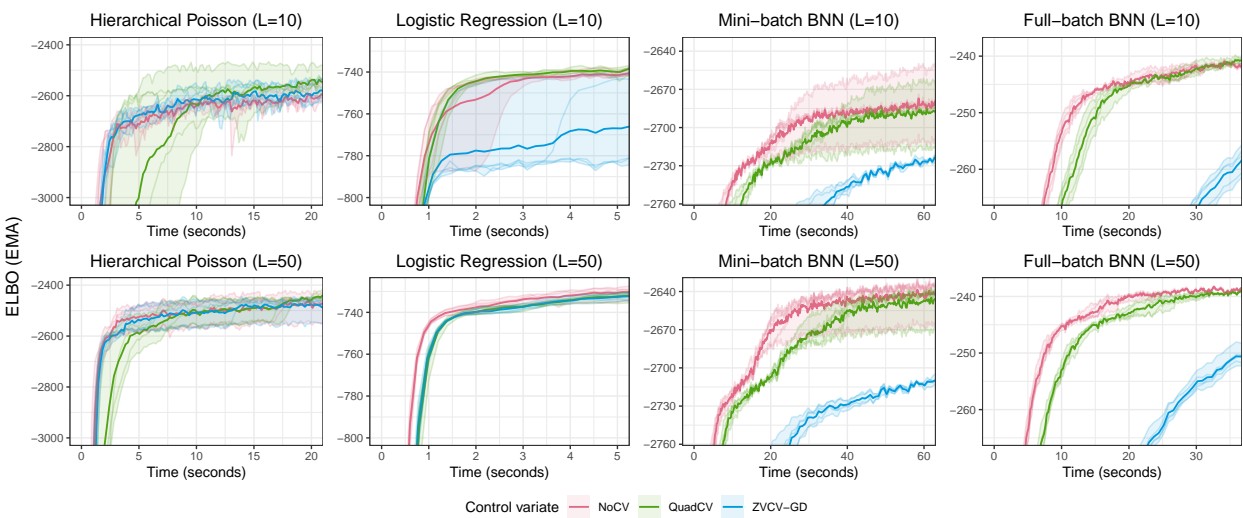

(b) Real NVP with 10 (top) and 50 (bottom) gradient samples.

Figure 8: ELBO is plotted against wall-clock time for different numbers of gradient samples $L$ and two families of $q$. The bold lines represent the mean of ELBO values recorded at the same iteration across five repetitions. The shaded area illustrates the range of ELBO values across five repetitions. The ELBO values are smoothed using an exponential moving average. A higher ELBO indicates better performance. See Figure 3 for plots where the bold lines represent the median ELBO.

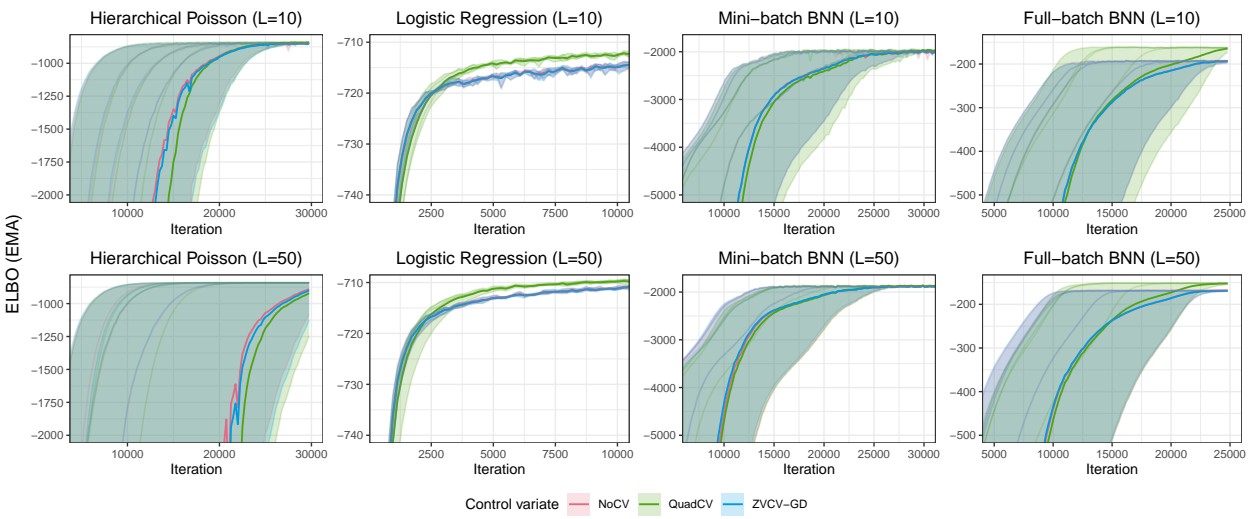

(a) ELBO versus iteration counts.

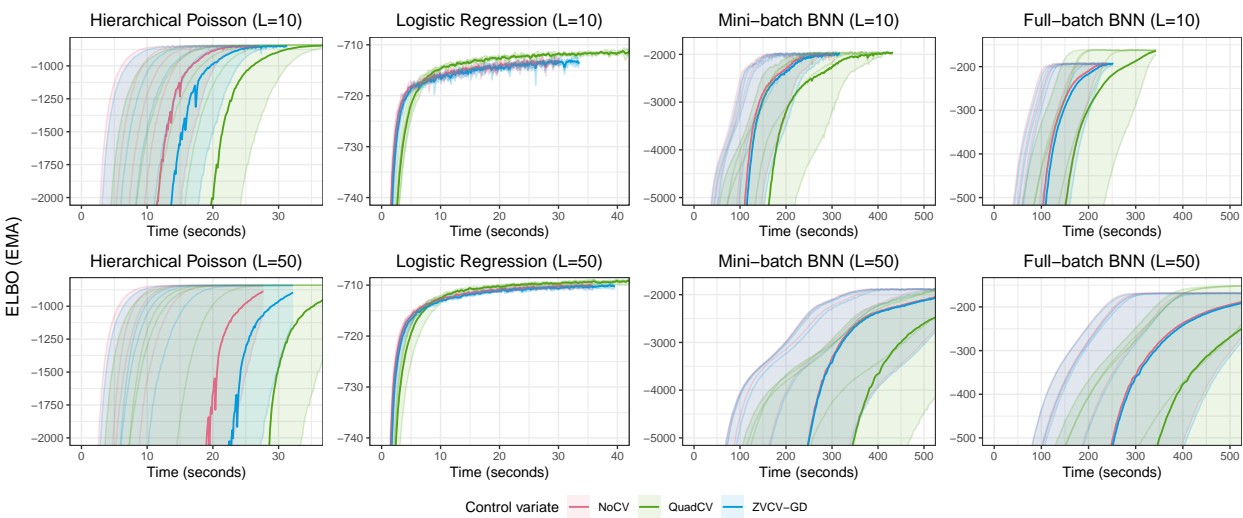

(b) ELBO versus wall-clock time.

Figure 9: ELBO is plotted against gradient descent steps and wall-clock time for varying numbers of gradient samples $L$ using rank-5 Gaussian. The bold lines represent the mean of ELBO values recorded at the same iteration across five repetitions. The ELBO values have been smoothed using an exponential moving average. A higher ELBO indicates better performance. See Figure 4 for plots where the bold lines represent the median ELBO.

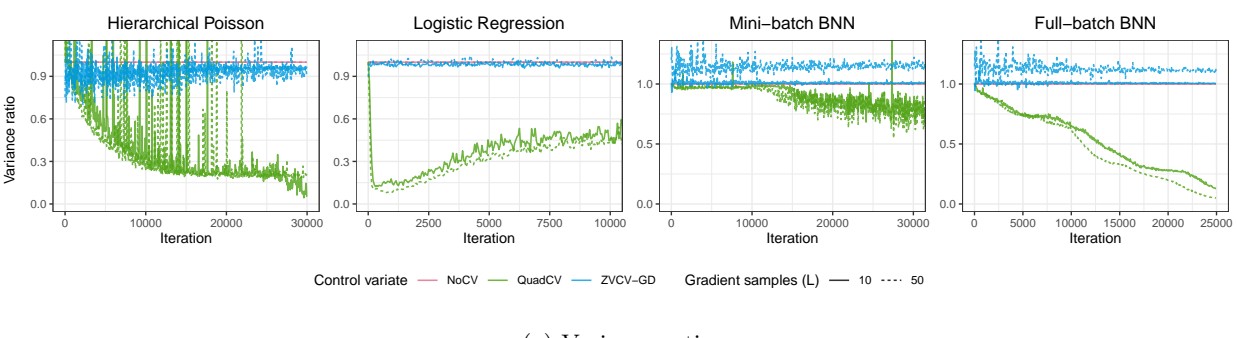

(a) Variance ratio

Figure 10: We present the variance ratio $\mathbb{V}[\hat{h}]/\mathbb{V}[\hat{g}]$ of rank-5 Gaussian, where $\hat{g}$ is NoCV and $\hat{h}$ is either ZVCV-GD or QuadCV, at each iteration. We show only the mean variance ratios recorded at the same iteration across five repetitions, omitting the individual variance ratios from each repetition to prevent clutter in the plots. Note that NoCV (in red) is always 1 by definition. We see that ZVCV-GD (in blue) struggles to reduce variance in the BNN models. There is also some overlap between $L = 10$ (solid green) and $L = 50$ (dotted green). A lower ratio indicates better performance. See Figure 5 for plots where the bold lines represent the median variance ratios.

# E   Individual runs of full-batch BNN with mean-field Gaussian

We zoom in on a particular model and variational family from the experiments in the main text. Our aim in this section is to look the trajectory according to each initialisation separately to help visualise the impact of initialisation on convergence. Due to space limitations, we have only included trajectories from full-batch BNN with mean-field Gaussian. In addition to the ELBO reported in the main text, we also report the downstream metric, log pointwise predictive density evaluated on a test set (test lppd), which is popular in the VI literature. Mathematically, the test lppd is defined as

$$\sum_{x \in \mathcal{D}_{\text{test}}} \log \left( |\mathcal{Z}|^{-1} \sum_{z \in \mathcal{Z}} p(x|z) \right).$$

Here, $\mathcal{D}_{\text{test}}$ represents a test set, $\mathcal{Z}$ is a set of samples drawn from $q(z; \lambda)$, and $|\mathcal{Z}|$ indicates the cardinality of $\mathcal{Z}$. We have set $|\mathcal{Z}| = 1000$ in our experiments. The test lppd is also referred to as the test log-likelihood, test log-predictive, or predictive log-likelihood in the literature (see, for example, Yao et al., 2019; Deshpande et al., 2022).

Figures 11a clearly show that trajectories vary substantially with different initialisations. This is consistent with the high variability of ELBO trajectories in Figure 1.

In all cases, increasing $L$, the number of gradient samples, effectively reduces the variance of the gradient estimator from the outset of the optimisation process. This stands in contrast to QuadCV, which only becomes effective after the quadratic approximation $\tilde{f}$ in (17) has been adequately trained (Figure 11b). Consequently, QuadCV performs poorly in the early and middle stages of optimisation (as seen in Repetitions 2 and 3 in Figure 11a).

Prior research on variance reduction in pathwise gradient estimators (e.g. Miller et al., 2017; Geffner & Domke, 2018; 2020) often aims to push the boundaries of attainable ELBO. Achieving this typically requires longer training periods. However, we are of the opinion that the additional ELBO gained through this effort does not warrant the extra computational cost incurred by implementing control variates. This is particularly relevant given that improvements in downstream metrics, such as test lppd, are marginal when compared to improvements achieved in the earlier stages of optimisation (note the y-axis scale in Figure 12a and 12b).

For instance, in Repetition 1, there is only a 3 nats improvement in test lppd (over a test set of size 100), while substantial improvements are observed in the earlier stages, often in the scale of hundreds. These 3 nats come at a cost of over 50% additional computation time compared to NoCV (as indicated in Figure 3a). Furthermore, it is worth noting that an improvement in ELBO does not invariably guarantee a substantial improvement in downstream statistics, as evidenced in previous works, such as Yao et al. (2018; 2019); Foong et al. (2020); Masegosa (2020); Deshpande et al. (2022).

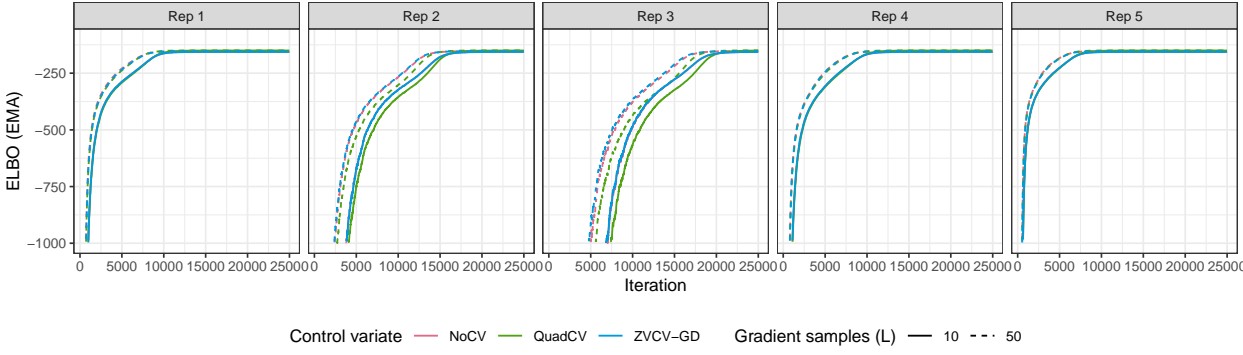

(a) ELBO trajectories. This is a zoomed-out version of the last column of Figure 1a. Higher values are preferred.

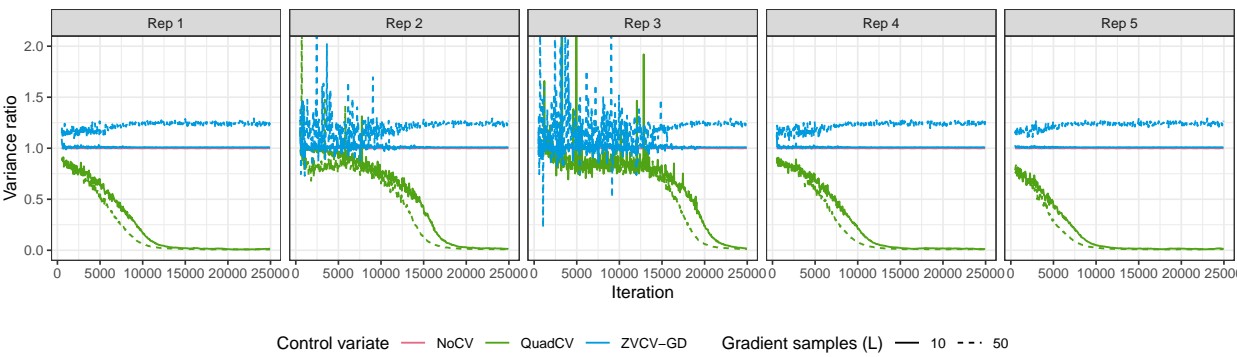

(b) Variance ratios. A reading of 1 indicates no variance reduction. Lower values are preferable.

Figure 11: The trajectories of ELBO and variance ratio for full-batch BNN with mean-field Gaussian are depicted over the course of iterations, with each of the five repetitions presented individually. By definition, the variance ratio of NoCV (red) is 1. Notably, there is a substantial overlap between NoCV (in red) and ZVCV-GD (in blue). In some cases, the distinctions between all three methods are hardly discernible. However, there is a relatively noticeable difference between $L = 10$ and $L = 50$.

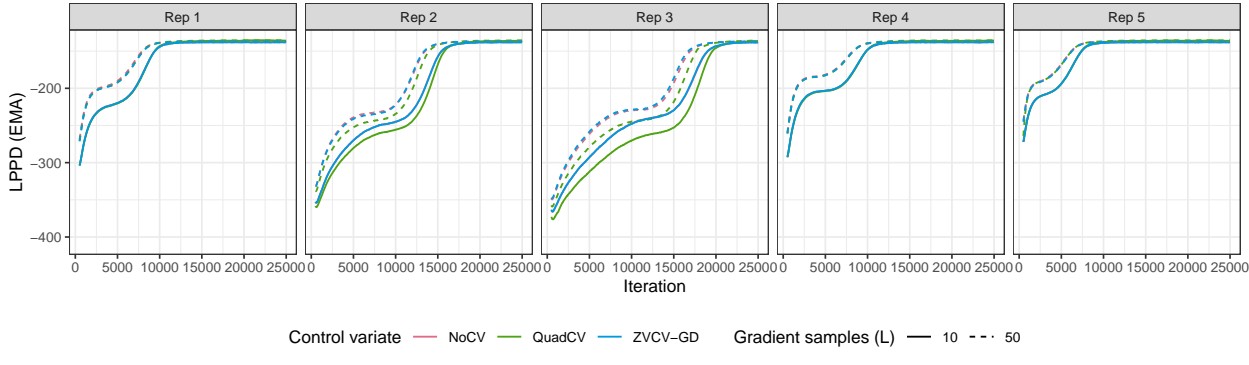

(a) Test lppd trajectories.

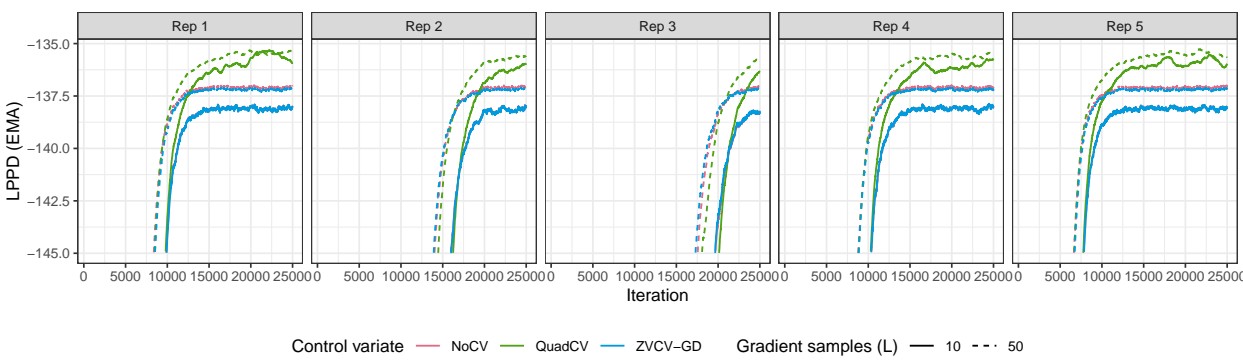

(b) Test lppd trajectories, zooming in between lppd $= (-145, -135)$.

Figure 12: The trajectories of test lppd for full-batch BNN with mean-field Gaussian are depicted over the course of iterations, with each of the five repetitions presented individually. Notably, there is a substantial overlap between NoCV (in red) and ZVCV-GD (in blue). In some cases, the distinctions between all three methods are hardly discernible. However, there is a relatively noticeable difference between $L = 10$ and $L = 50$. Higher values are preferred.

## F   Comparison of ZVCV-GD with different hyperparameters

We conducted experiments with ZVCV-GD that explore various hyperparameter settings, running with both first- and second-order polynomials (Figure 13), and testing different number of steps in the inner gradient descent loop (Figure 14). We focus on the hierarchical Poisson model using a mean-field Gaussian and setting $L = 10$. We repeated the experiment five times, each time with different initialisations. The red trajectories in Figure 13 and 14 correspond to the default settings of ZVCV-GD as specified in Section 6.

Figure 13b reveals that second-order ZVCV-GD did not effectively reduce variance in the gradient estimator; instead, it introduced additional noise into the estimator. This detrimental impact is also evident in the ELBO trajectories, as shown in Figure 13a. In light of these findings, we concluded that the simpler first-order ZVCV-GD is preferable over the second-order variant.

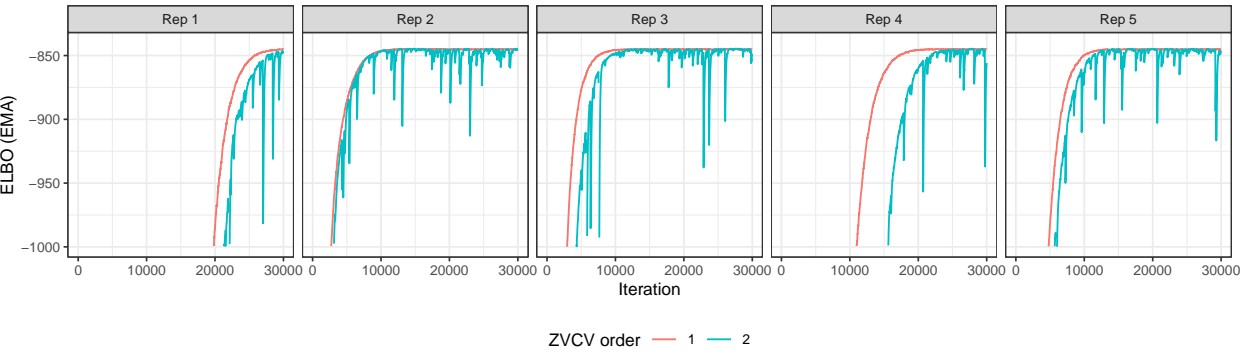

(a) ELBO trajectories. Higher values are preferable.

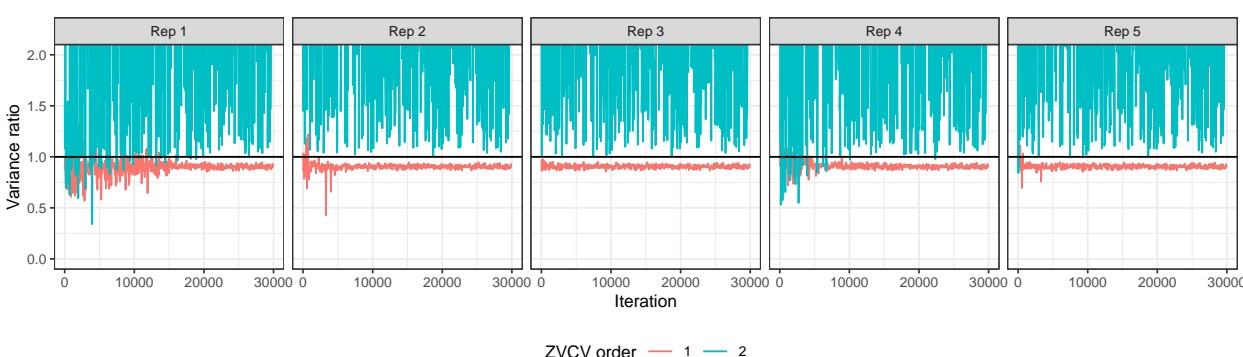

(b) Variance ratios. A reading of 1 indicates no variance reduction. Lower values are preferable.

Figure 13: ELBO trajectories and variance ratios for hierarchical Poisson models using mean-field Gaussian, ZVCV-GD with $L = 10$, and first- and second-order ZVCV-GD both with 4 inner GD steps. The experiment was repeated five times, each time with different initialisations.

In Figure 14, we present the ELBO trajectories and variance ratios obtained by running the inner gradient descent (GD) of ZVCV-GD with three different settings: 4 steps, 20 steps, and 'until convergence'. Here, 'convergence' is defined as the point at which the residual of the inner least squares problem in (13) no longer decreases substantially.

We observe that running the inner GD until convergence does not necessarily yield the greatest variance reduction, as illustrated in Figure 14b. This phenomenon can be attributed to overfitting the linear regression in (13), where the number of rows in $C$ is considerably smaller than the number of columns. On the other hand, iterating the inner GD 20 times achieves a more substantial variance reduction compared to the default 4 steps.

However, it is worth highlighting that there is no discernible impact on the ELBO trajectories when varying the number of GD steps, as demonstrated in Figure 14a.

The optimal number of steps is not always evident without experimentation. Hence, we typically opt for 4 steps to balance computational efficiency and the risk of over-optimizing the inner GD process.

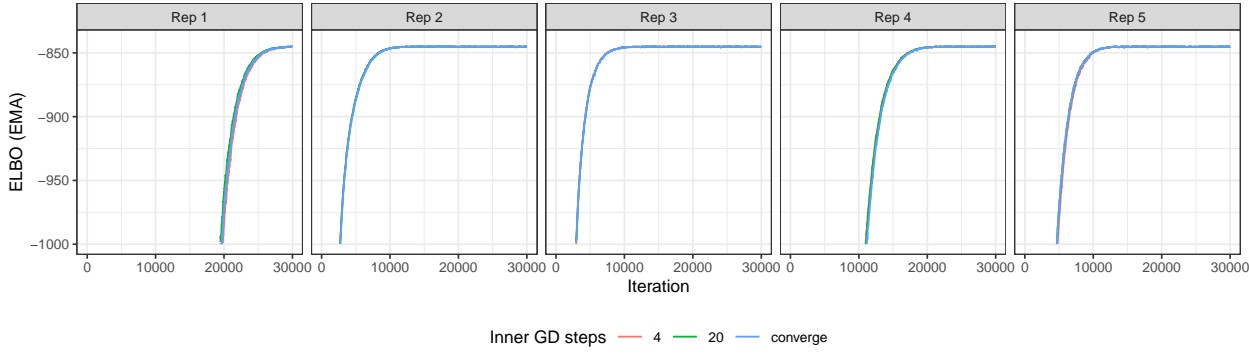

(a) ELBO trajectories. Higher values are preferable.

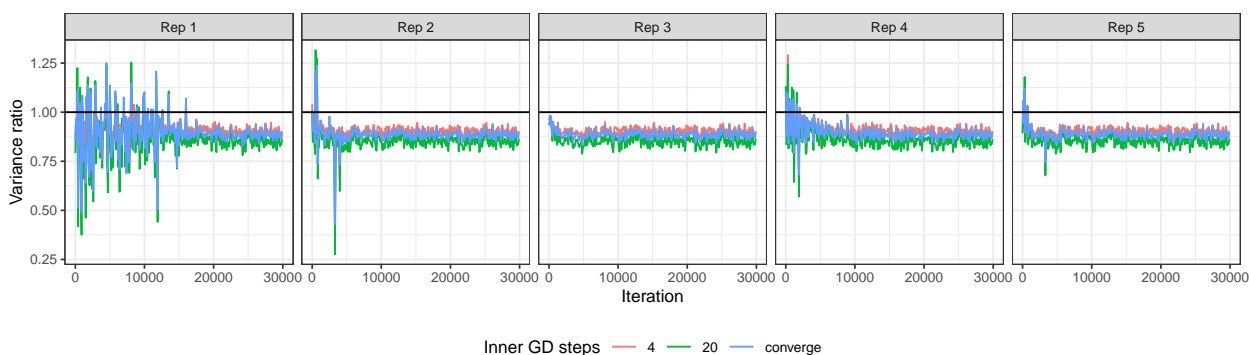

(b) Variance ratios. A reading of 1 indicates no variance reduction. Lower values are preferable.

Figure 14: ELBO trajectories and variance ratios for hierarchical Poisson models using mean-field Gaussian, (first-order) ZVCV-GD with $L = 10$, running with different number of steps in the inner gradient descent. The experiment was repeated five times, each time with different initialisations. The ELBO trajectories for different GD steps are practically indistinguishable. The erratic variance ratio readings occur during the early optimisation stages in the low ELBO region, where gradient magnitudes are substantial.

