# OpenReview forum: "Pathwise gradient variance reduction with control variates in variational inference"
_TMLR — Rejected by TMLR_

### Review · Reviewer_wCDN · 2023-09-30

**Summary Of Contributions:**

The authors (probably) aim to demonstrate the application of the zero-variance control variates, initially proposed to reduce variance in pathwise (reparameterized) gradients, especially in scenarios where determining the variational parameters during training is challenging (e.g., normalizing flow). This paper provides comprehensive explanations and reviews of existing studies (from Section 1 to Section 5.1).
In Sections 5.2 and 5.3, they introduce an approximation-based variance reduction method, which minimizes the penalized least square loss using gradient descent based on the similar approach in Si et al. (2022). Empirical validation of the proposed method is conducted in Section 6 across three variational families: Mean-field Gaussian, Rank-5 Gaussian, and Real NVP.

**Audience:**

Yes

**Broader Impact Concerns:**

I believe that this work does not raise any ethical concerns because it is a methodological study focused on gradient estimation with variance reduction.

**Claims And Evidence:**

No

**Requested Changes:**

- It would be greatly appreciated if you could clarify the positioning of this paper. In other words, is this intended to be a review paper on pathwise gradient variance reduction methods, or is the main contribution centered around the methods proposed in Section 5.2.2? Please provide explicit clarification on this point.
- It would be advisable to check the implementation and environment of the experiments for any issues and reconsider the evaluation methods. At least to me, there seems to be an unnatural discrepancy in the results of repeated experiments. If there are no issues, it would be beneficial to investigate why this phenomenon is occurring and provide an explanation for it in Section 6. This aspect is closely related to one of the main acceptance criteria of TMLR: *a discussion based on accurate evidence.*
- Please enhance the overall quality of the paper's presentation. Mistakes in citations and inconsistencies in descriptions may reduce the reader's interest.

**Strengths And Weaknesses:**

First and foremost, I would like to express my sincere respect for all the efforts the authors have invested in this paper.
Furthermore, it should be noted that the following review is explicitly written with an understanding that TMLR places importance on *accuracy, convincing, and clear evidence,* as well as *capturing readers' interest* over novelty and impact.

# Strengths
- The authors have focused on an intriguing problem, which is that the reparameterized gradient estimator cannot be applied to variational inference scenarios where parameters cannot be traced, such as in the case of normalizing flows. This issue has the potential to become an interesting research topic.
- This paper provides a detailed review of existing research on pathwise gradient estimation from Section 1 to Section 5.1.

# Weakness
## Concerns for discussion based on accurate evidence
- Since this is a methodological paper, it is crucial to provide accurate and convincible empirical evidence to assure the validity of the contribution. However, I have concerns about the credibility of the experimental results. I raise several points of concern as follows.
  - **Figure 1, Overall results**: For each method, there appears to be a large discrepancy in the results obtained by repeating the experiment five times. For example, when examining the simplest experimental setting (the top-left figure; Hierarchical Poisson (L=10)), it is observed that the convergence of the method without using CV varies greatly (in some cases, it converges at $8000$ iterations, while in others, it has not converged even after $30,000$ iterations.). This extreme divergence seems odd if the experiment is repeated for the same model and hyperparameter settings under a simple experimental setup. Additionally, in cases like *Mini-batch BNN (L=50)*, the ELBO indicated by the red line suddenly drops around iteration $15,000$. In my opinion, these behaviors are highly unusual, and it seems unlikely that they would occur in an appropriately controlled experimental environment. Similar irregularities can be observed sporadically in other experimental settings as well.
  -  **Figure 1, regarding the "median" ELBO among the 5 repetitions**: It is unclear whether this *median* refers to the median calculated from the numerical ELBO values recorded at the same iteration across the 5 repetitions, or if it means choosing the results from the third best experimental results out of the 5. It appears to be the latter from Figure 1. This choice of adopting the *third-best* experimental results raises questions about the rationale behind such a selection. Without a valid explanation for why the *third-best* experimental results were chosen, the interpretability of the evaluation method seems to be compromised. Furthermore, this evaluation approach leads to cases where methods are unfairly evaluated. For instance, in the experimental results for *Logistic regression (L=10)* and similar cases, the proposed method appears to perform competitively with existing methods, yet the bold line is placed considerably lower. It is difficult for me to adopt as accurate and convincing evidence the results of an experiment evaluated in a way that seems inappropriate. Considering that the discussion primarily revolves around the gradient variance due to i.i.d. Monte Carlo samples, it would indeed be more appropriate to evaluate performance using the mean ± std. of the ELBO.
   - **Figure 1, the top-right figure**: It appears that the experimental results for *No CV* are missing from the figure (red line).

## Concerns regarding the interest of TMLR readers

- To capture the readers' interest, readability is one of the crucial points. However, in this paper, it appears that the following elements are having a negative impact on readability.
  - **Position of this paper**:  If this paper is positioned as a review paper on pathwise gradient variance reduction, then the detailed review of existing studies extending up to Section 5.1 would indeed contribute to researchers and students interested in the relevant field.　However, if the algorithm proposed in Section 5.2.2 is to be considered an essential contribution, its explanation seems inadequate. In the current structure, the evaluation of this paper may vary depending on which perspective is taken. Therefore, it seems that the authors should first clarify the main claim they want to convey in this paper.
  - **Discussion of related studies**: In this paper, it is assumed that in problem settings where the variational distribution during training cannot be obtained exactly, as in the case of Real NVP, the variance in gradient estimation negatively affects convergence and predictive performance. However, there seems to be a lack of discussion regarding related studies on this aspect, leaving the paper's position unclear. Are there any related work around this issue? Given that this aspect appears to be of significant interest to TMLR readers, providing adequate discussion on it would likely make the paper even stronger.
  - **Citation deficiencies etc.**: There are inconsistencies throughout the paper in terms of how references are presented and the use of abbreviations, which significantly impairs the overall readability and raises concerns about detracting from the interests of the intended readers. For instance, there is a lack of reference information for [Belomestny et al. (2018)]. Furthermore, although abbreviations such as *CV* and *VI* are introduced, the paper later inconsistently reverts to using expressions like *control variates* and *variational inference.*

---

> ### Author Response · Authors · 2023-10-17
> **Addressing comments from Reviewer wCDN**
>
> ### **Figure 1, observed discrepancies across experiment repetitions in overall results**
>
> The experiments were designed to demonstrate a) the robustness of each method to different initialisation, and b) the convergence time of each method. This is in contrast to Geffner’s which only ran their experiments with the same initialisation. Based on our observation (Appendix D), initialisation is perhaps having the biggest influence on the ELBO trajectories relative to other factors (e.g. variance in the gradient estimator).
>
> In other words, it is to be expected that there is variation in the results when we repeat the experiment five times. Regarding the extreme divergence observed in BNNs with real NVP, thank you for pointing this out. After revisiting our implementation, it seems that we were using a learning rate that was too large. We have since tuned down the learning rate and rectified the issue.
>
>
> ### **Figure 1, regarding the "median" ELBO among the 5 repetitions**
>
> The median ELBO refers to the median computed from the numerical ELBO values recorded at the same iteration across the 5 repetitions. We are not adopting the “third-best” experimental results. This is now clarified in the paper, and we apologise for not making this clear previously. We agree that interpreting the “third-best” result is not sensible.
>
> We agree with the reviewer’s suggestion that since the interest is in gradient variance due to Monte Carlo samples, it would be more appropriate to report mean plus/minus std. However, we choose to report the median, rather than mean, of the ELBO value due to the variability of the ELBO primarily coming from initialisation rather than the gradient estimator. In our opinion, median ELBO gives a more honest assessment on the merit of each methods. For example, a method with 4 good runs and 1 horrible run might compare unfavourably to a competing method with 5 mediocre runs when looking at their mean ELBO trajectory, yet the former is more likely to be the desirable method. We have also included the trajectory of each individual run in the new Appendix D to provide a clearer picture.
>
> ### **Figure 1, the top-right figure**
>
> Thanks for pointing this out. There is a significant overlap between NoCV and ZVCV-GD. We have added a comment to make this clear in the figure caption.
>
> ### **Position of this paper**
>
> We agree that we did not make our position very clear in the previous submission. We have since clarified our position that this work is a review of control-variate-adjusted pathwise gradient estimators for variational inference. The introduction of ZVCV in this context can be viewed as an exercise in completeness. Namely, the existing control variates for pathwise gradient estimators cannot handle the setting where the variational distribution does not have closed-form mean and covariance.
>
> ### **Discussion of related studies**
>
> It is more accurate to say that, in this paper, it is assumed that the variance of the pathwise gradient estimator in variational inference can negatively affect convergence time, robustness to poor intialisation, and the final ELBO achieved. We conjecture this can happen for variational distributions as simple as mean-field Gaussian to more complex distributions such as normalizing flows. The proposed ZVCV simply fills the gap in the literature for variational distributions without closed-form mean and covariance.
>
> As for related works on the adverse effect of a high-variance gradient estimator in VI, there is a plethora of literature albeit for the REINFORCE gradient. It seems that the adverse effect for pathwise gradients in VI is an under-studied problem which this work seeks to address.
>
> ### **Citation deficiencies etc.**
> Thank you for pointing out the inconsistencies. We have fixed all the citation deficiencies, including Belomestny et al. 2018, and inconsistency in abbreviations.
>
> ### **Requested changes**
> We have addressed the requested changes, please see comments above and the general comment to all reviewers.

---

> > ### Comment · Reviewer_wCDN · 2023-10-26
> > **Acknowledgment of the author's responses and further discussion (1)**
> >
> > First and foremost, I would like to express my sincere appreciation for the time and effort the authors have dedicated to responding to my review.
> > I've read the response and the reviews provided by other reviewers.
> > Although the authors clarified the position of the paper, I still have concerns about the novelty and validity of the claim.
> > I summarized my further comments and questions as follows.
> > I would like to decide whether to recommend acceptance of this paper, considering the authors' responses to these.
> >
> > ## Gradient variance and initialization
> > - I believe that a more detailed explanation of the experimental design is necessary. Upon re-reading the revised manuscript, I couldn't find a clear description of how randomized the initial values for various hyperparameters are.
> > - In this paper, there are numerous hyperparameters, such as variational parameters like $\lambda$, prior parameters, learning rates, and control variate weight $\beta$. While the authors claim that initialization has the most significant impact on the ELBO during testing, it seems essential and beneficial to determine which of these parameters actually had the most negative influence (e.g., conducting experiments under various settings of $\beta$ or learning rate and exploring the relationship based on figures like "test Performance vs. $\beta$" could be one of a useful approach).
> >
> > - When training a standard Neural Network (NN) with SGD, it is conceivable that even when taking sufficiently large mini-batches to reduce the variance in stochastic gradients, predictive accuracy can deteriorate if inappropriate learning rates are set. This phenomenon can also occur in Bayesian NN trained by using VI. Then, how does this fact differ fundamentally from the claim in this paper that *inappropriate initial value settings have a more significant impact on the performance of VI than gradient variance*?
> >
> > - The detrimental consequences of improper initialization are widely acknowledged as a significant concern in machine learning, extending beyond VI with variance reduction. Within the realm of Bayesian machine learning, various studies have addressed this issue, including seeking suitable priors [1] and exploring initialization strategies [2]. In my view, these topics should be regarded as distinct from the investigation of variance reduction effects, as the former represents a more overarching concern compared to the study of variance reduction.
> >
> > - The theoretical analysis regarding the effectiveness of variance reduction using reparametrization gradients often focuses on the convergence of gradients, for example, in [3]. However, these analyses merely state that convergence to (local) optima is improved by variance reduction and does not necessarily guarantee better predictive accuracy of the parameters after convergence. In the context of variance reduction, regardless of the quality of the initial values, the primary focus lies in devising variance reduction techniques that achieve better convergence properties in the model and inference algorithms. Therefore, in studies of variance reduction in the vein of Geffner and others in the context of variational inference, it is common to experimentally confirm whether the convergence properties have improved on average concerning the randomness of initial values and datasets. In my opinion, the issue of the impact of gradient variance on convergence, regardless of the quality of the initial values, and the issue of poor initial values promoting convergence to bad local minima are separate matters. The former focuses on convergence, while the latter concentrates on predictive performance. Hence, comparing them side by side may seem somewhat peculiar.
> >
> > - If the numerical experiments in this paper are appropriately performed, the claim that initial value setting may have a more detrimental effect than gradient variance is consistent to some extent with my conjecture that the effect of inappropriate initial value is a more global problem than gradient variance. Since initialization strategies to achieve better prediction accuracy have been studied in recent years, it should be taken into account that these may be avoided by setting initial values with reference to [1,2] and others.
> >
> > (Due to character limitations, our reply will be continued in the next thread.)

---

> > > ### Comment · Reviewer_wCDN · 2023-10-26
> > > **Acknowledgment of the author's responses and further discussion (2)**
> > >
> > > # Using median ELBO
> > > - I still have reservations regarding the assessment of performance using the median. If this paper aims to evaluate the sensitivity to initial configurations experimentally, then when we have one method that performs well in four out of five situations but poorly in one, and another method that performs slightly better in all five situations but not as well as the former in one, shouldn't the latter be considered the method with better robustness in terms of initial value settings? Assessing a method with a robust metric like the median corresponds to appraising performance while, to some extent, disregarding unfavorable outcomes. This approach may unfairly favor default-sensitive methods and deviate from the original intent of the evaluation.
> > >
> > > # Position of this paper & Discussion with related studies
> > > - I understand this paper was crafted as a review paper for control-variate-adjusted pathwise gradient estimators. It would be best for the authors to explicitly convey this purpose in the title and abstract of this paper.
> > > - If the authors intend to delve into the instability associated with initial values, it seems crucial to include a section that cites and discusses studies related to initialization in the context of VI, BNN, etc., such as [2].
> > >
> > > ## Citation
> > > [1]: V. Fortuin et al. Bayesian Neural Network Priors Revisited. ICLR2022.
> > > https://arxiv.org/pdf/2102.06571.pdf
> > >
> > > [2]: S. Rossi et al. Good Initializations of Variational Bayes for Deep Models. ICML2019.
> > > http://proceedings.mlr.press/v97/rossi19a/rossi19a.pdf
> > >
> > > [3]: K. Kim et al. On the Convergence and Scale Parameterizations of Black-Box Variational Inference. NeurIPS2023. (This paper has been accepted recently, but the arXiv version has been uploaded since 24 May 2023.)
> > > https://arxiv.org/pdf/2305.15349.pdf

---

> > > > ### Author Response · Authors · 2023-10-31
> > > > **Further discussion on using median ELBO, position of this paper and related studies**
> > > >
> > > > ### **Using Median ELBO**
> > > >
> > > > We agree that using the median can unfairly favor default-sensitive methods. As a response, we have recreated all the figures in Section 6 and Appendix C and replaced the medians with means. These 'mean-version' plots are included in Appendix D (colored in blue). We believe the ideal approach would be examining each run separately, as shown in Appendix E, but this would consume a significant amount of space. As a compromise, we shaded the area in both the mean and median plots to illustrate the range of ELBO values, although this does result in somewhat cluttered plots.
> > > >
> > > > ### **Position of the paper**
> > > >
> > > > In response to the reviewer's request, we have revised the title to '*Pathwise gradient variance reduction with control variates in variational inferences*' and updated the abstract to emphasize that the primary focus of this work is a review of control-variate-adjusted pathwise gradient estimators.
> > > >
> > > > ### **Related studies**
> > > >
> > > > We apologize for any confusion resulting from our exposition and experiments related to the initialization of $\lambda$. We have removed the reference suggesting that variance reduction of the pathwise gradient is a direct solution to addressing bad initialization of $\lambda$ in VI. It was not our intention to study the instability associated with initial values.

---

> > > ### Author Response · Authors · 2023-10-31
> > > **Further discussion on gradient variance and initialization (1)**
> > >
> > > We would like to express our gratitude to the reviewer for providing such detailed feedback. We have made the necessary revisions which are highlighted in blue. A common theme in the second round of comments revolves around the initialization of $\lambda$. It has prompted us to reconsider why we specifically focused on evaluating the robustness of control-variates-adjusted pathwise gradient estimators in VI to poor $\lambda$ initialization, as there are other hyperparameters for which we don't assess robustness. For those hyperparameters, we perform parameter sweeps to determine suitable values. Similarly, we could adopt a similar approach to explore different initialization strategies to find an effective one.
> > >
> > > After sitting with this for some time, we agree with the reviewer's suggestions and have removed the discussion on evaluating the effectiveness of control variates in terms of robustness against poor $\lambda$ initializations. However, we have retained the results for different $\lambda$ initializations, aligning with the spirit of the reviewer's comment: *“… in studies of variance reduction in the vein of Geffner and others in the context of variational inference, it is common to experimentally confirm whether the convergence properties have improved on average concerning the randomness of initial values and datasets.”*
> > >
> > > We hope that these changes address your concerns. If there are any points that remain unclear or require further revision, please don't hesitate to inform us. There is still some time before the deadline, and we are committed to making additional improvements if needed.
> > >
> > > ### **Description of hyperparameters**
> > >
> > > Thank you for your careful re-reading of the revision. Below, we summarise the hyperparameters involved in each of the three methods (QuadCV, ZVCV-GD, NoCV) and explain how we determined the hyperparameter values in the experiments. We have also reformatted the introduction of Section 6 and consolidated all hyperparameter settings here.
> > >
> > > - QuadCV (Algorithm 1): the outer loop learning rate $\gamma^{(\lambda)}$m the inner loop learning rate $\gamma^{(v)}$. The term 'inner loop' may appear unusual because we are taking a single descent step in $v$, but this terminology is appropriate because, in its most general form, QuadCV involves an inner loop gradient descent to learn $\tilde f$. For $\gamma^{(\lambda)}$, we conducted a sweep using values from the set $\{0.1, 0.01, 0.001, 0.0001\}$ and selected the one that led to the fastest convergence to a respectable ELBO. (We omitted this detail in the initial manuscript, but this has been added in Section 6 of our second revision and highlighted in blue.) As for $\gamma^{(v)}$, we opted not to sweep it; instead, we set it to a fixed value following the approach of the original work.
> > > - ZVCV-GD (Algorithm 2): the outer loop learning rate $\gamma^{(\lambda)}$, the inner loop learning rate $\gamma^{(\alpha,\beta)}$, the number of inner loop steps. As in Algorithm 1, we did a sweep of $\gamma^{(\lambda)}$ from the set $\{0.1, 0.01, 0.001, 0.0001\}$. In contrast to Algorithm 1, we also sweep the inner loop learning rate from the same set. The reviewer also asked about the control variate weight $\beta$. We assume they were referring to $\beta_0$ in Algorithm 2 — this is always at zero for a specific reason and hence is not a hyperparameter. Likewise $\alpha_0$ in Algorithm 2 is not to be viewed as a hyperparameter. As shown in Appendix E, we conducted a sweep to explore the optimal number of steps for the inner loop.
> > > - NoCV: the outer loop learning rate $\gamma^{(\lambda)}$, selecting from the set $\{0.1, 0.01, 0.001, 0.0001\}$.
> > >
> > > The choice of prior parameters is a modeling decision, and we closely follow the setup in Miller et al. 2017 and Geffner et al. 2020 to ensure an apples-to-apples comparison.
> > >
> > > ### **Examining which hyperparameters has the most influence**
> > >
> > > The initialization of the variational parameter $\lambda$ is expected to have an impact on the performance of VI. In our experiments, our original goal was to assess whether control-variate-adjusted pathwise gradient estimators in VI are more robust to poor initializations of $\lambda$. We also note that the initialization of other hyperparameters, such as learning rates, will also influence the performance of VI. (Refer to the previous explanation for why we don't consider prior parameters and the control variate weight, $\beta$, as hyperparameters.) In fact, there are numerous other factors to consider, such as minibatch size and general optimizer hyperparameters. In theory, it's possible to empirically evaluate whether control-variate-adjusted pathwise gradient estimators in VI are more robust to unfavorable batch sizes, suboptimal optimizer hyperparameters, and so on. However, we acknowledge that this would be a separate line of research. In our new revision, we primarily focus on convergence performance and, secondarily, on downstream predictive accuracy.
> > >
> > > (cont.)

---

> > > > ### Comment · Reviewer_wCDN · 2023-11-03
> > > > **Acknowledgement and finalization of review**
> > > >
> > > > Thank you for your sincere response to my comments. I have reviewed the revisions made.
> > > >
> > > > While there are still concerns, providing further feedback or discussion on the issues with this paper would likely be considered an over-contribution as a reviewer. Additionally, with the deadline approaching, it might be challenging to provide specific comments. Therefore, I will convey my current opinions on this paper to the editor.
> > > >
> > > > One final piece I can offer is to clarify why you decided to write this review paper including the reason why you focused on the variance reduction method only for the pathwise gradient, its significance, and what specific problems it aims to address. This can be a helpful step in strengthening the paper.
> > > >
> > > > With due respect to the time and effort put into this paper by the authors,
> > > > reviewer wCDN.

---

> > > ### Author Response · Authors · 2023-10-31
> > > **Further discussion on gradient variance and initialization (2)**
> > >
> > > ### **Inappropriate initial value settings having more impact on the performance than gradient variance**
> > >
> > > We apologize for giving the impression of making the claim that inappropriate initial values settings have a more significant impact than gradient variance. While we do assert that the inappropriate initialization of $\lambda$ significantly affects the performance of VI, and high gradient variance also has a significant impact, we do not claim that the former's impact is greater than the latter's.
> > >
> > > ### **Peculiar to compare initialisation and variance reduction side by side**
> > >
> > > We absolutely agree that initialization strategies in VI or more generally, Bayesian machine learning, is a separate line of investigation than variance reduction of pathwise gradient estimators. In our new revision, our primary focus remains on convergence performance, with secondary attention given to downstream predictive accuracy.
> > >
> > > Following the reviewer’s suggestions, our interpretation of the experimental results in the new revision largely rests on the convergence properties of the control-variate-adjusted pathwise gradient estimator, irrespective of the initial value of $\lambda$. We are repeating the experiment to average out the impact of different initial values of $\lambda$.
> > >
> > > ### **Taking into account recent research in initialisation strategy**
> > >
> > > We agree that initial values can indeed have a substantial impact on convergence performance and predictive accuracy. However, our motivation of repeating the experiment is to average out the impact of different initial $\lambda$ values to give a more honest assessment of each control variate on convergence.

---

### Review · Reviewer_hVtQ · 2023-10-04

**Summary Of Contributions:**

This work proposes a new variance reduction technique for variational inference with the reparameterization trick. It applies zero-variance control variates (ZVCV) constructed by Stein operators to reduce the variance of estimating the expectation. This control variates do not require to know the analytical form of the mean and covariance of the variational distribution. However, the empirical results showed that the proposed gradient estimator did not provide substantial variance reduction. Instead, it is found that increasing the number of gradient samples is a more cost-effective method for reducing variance.

**Audience:**

No

**Claims And Evidence:**

Yes

**Requested Changes:**

Since my concern is more about the methodology itself, I would suggest the authors to revisit their proposed approach and try to improve upon the few points raise above.

**Strengths And Weaknesses:**

## Strengths

* This paper is clearly written and has sufficient discussion of closely related work.
* The use of Langevin Stein operator to avoid the difficulty of computing mean and variance of normalizing flow type of variational distribution is well-motivated. This is a major weakness of the Geffner et al. (2020) approach and is correctly identified in this work.

## Weaknesses

* Only using first-order polynomials in the control variate construction feels rather limited. The fact that the control variate falls back to a linear form is also quite unsatisfying. This means the control variates could not be very effective when the gradient of log likelihood term is highly nonlinear.
* Although it can be argued that in large data problems, the posterior will become more Gaussian and the linear control variate can potentially be effective, it does not justify the sophisticated derivation to obtain such a CV. Could we apply standard regression based linear control variates instead? I assume the results can be very similar.
* The way the coefficient before the control variates is estimated is also concerning. It requires a inner loop of gradient descent to solve an optimization problem, mainly because the exact solution is computationally expensive. However, the inner loop is only iterated 4 times and it is unclear the coefficients obtained in this way will be close to the exact solution in the end.
* The empirical results are negative. This is not to say that negative results are not valued. But, the negative result can be sort of predicted from the reasons listed above--the control variates being too simple.

---

> ### Author Response · Authors · 2023-10-17
> **Addressing comments from Reviewer hVtQ**
>
> ### **Only using first-order polynomials feels rather limited.**
>
> We have included results on second-order ZVCV-GD in Appendix E. These results show that second-order ZVCV-GD can sometimes have a detrimental effect. We suspect that this may be attributed to the inherent challenge of solving an underdetermined linear system with a large number of variables, all while avoiding overfitting $C \beta$ to the gradient samples.
>
> ### **The posterior will become more Gaussian and the linear control variate can potentially be effective.**
> The motivation for the proposed ZVCV is not based on the posterior becoming more Gaussian asymptotically. In fact, we know the posterior distribution over neural network weights in BNNs looks far from Gaussian (Wei et al 2022). Note that we are already using a linear combination of control variates and using standard regression to learn the optimal coefficients of this linear combination.
>
> Wei, S., Murfet, D., Gong, M., Li, H., Gell-Redman, J., & Quella, T. (2022). Deep learning is singular, and that’s good. *IEEE Transactions on Neural Networks and Learning Systems*.
>
> ### **The inner loop is only iterated 4 times. Unclear if the coefficients will be close to the exact solution.**
> We have included additional experiments in Appendix E to investigate the impact of different descent steps (4, 20, and until convergence). It is worth noting that iterating four times may not always be the optimal choice, but it effectively reduces a substantial portion of the variance compared to 20 iterations. Notably, extending the inner gradient descent until convergence does not necessarily yield the most significant reduction in variance, as Eq 13 represents an underdetermined system. Prolonged iterations in the inner gradient descent run the risk of overfitting $C \beta$ to the gradient samples.
>
> ### **Negative results due to the control variates being too simple.**
> First-order ZVCV is admittedly simple in the setting of Gaussian base distribution. Interestingly, we saw no improvements by using second-order ZVCV, which give less simplistic control variates than first-order ZVCV. Also notably, QuadCV is not a simple control variate, yet its impact on convergence remains unimpressive. Therefore it would seem that the control variates being too simple is not the primary reason for the negative result here.

---

> > ### Comment · Reviewer_hVtQ · 2023-11-03
> > **Response**
> >
> > Thanks to the authors for the detailed response. Please find my comment to each individual point below.
> >
> > > We have included results on second-order ZVCV-GD in Appendix E.
> >
> > I assume you mean appendix F. I appreciate the additional experiments. However, I believe the paper (if repositioned as a review paper) can be much stronger if it goes beyond just showing the results and provides insights on why some of these methods work better than others. For example, it would be great to design such an experiment to test the hypothesis you mentioned: "inherent challenge of solving an underdetermined linear system with a large number of variables, all while avoiding overfitting to the gradient samples."
> >
> > > Note that we are already using a linear combination of control variates and using standard regression to learn the optimal coefficients
> >
> > This is exactly my point. If this is the case, I believe the whole Stein control variates construction is unnecessary and complicates the presentation.
> >
> > > Also notably, QuadCV is not a simple control variate, yet its impact on convergence remains unimpressive. Therefore it would seem that the control variates being too simple is not the primary reason for the negative result here.
> >
> > I don't think the justification here (based on QuadCV) is relevant. It's unimpressive, but still significantly better than other estimators you compared.

---

### Review · Reviewer_CLe6 · 2023-10-06

**Summary Of Contributions:**

The paper proposes a new control variates method to reduce the variance of pathwise (reparametrization) gradient in the context of variational inference. The new methods reduces the requirement of variational distribution by the existing methods targeting the same problem.

**Audience:**

Yes

**Claims And Evidence:**

Yes

**Requested Changes:**

- Reflect the empirical advantages of the proposed method in the simulations.
- Improve the writing clarity and accuracy.

**Strengths And Weaknesses:**

Strengths:

- The paper contains a clear review of the related literature.
- It identifies the limitation of existing methods.
- It provides a fair discussion of the experimental results.

Weakness：

- One major concern is the experimental results. In the main results of Figure 1, the proposed ZVCV-GD does not achieve the optimal objective value in any case. It raises the question of its effectiveness.

- Based on the observation, the paper discusses that "dramatic reduction in gradient variance can fail to deliver any discernible effect on the ELBO",  "reducing the gradient variance is insufficient to improving downstream performance," and "we found that increasing the number of gradient samples is a highly cost-effective method for reducing variance." The reviewer holds similar questions about whether it makes a big difference to reduce the variance of reparameterization gradient, and whether the variance reduction methods can be more efficient than taking multiple Monte-Carlo samples. Since the practitioners often take only a single sample estimate for the reparameterization gradient, it might hint at the negative results concluded by the paper. Unfortunately, the negative results may not serve as enough reasons to support the acceptance.

- I do hold some concerns about the negative results. First, in Figure 1, Quadratic CV does achieve the best performance, and in Figure 2, Quadratic CV has the lowest ratio. This seems to suggest the variance reduction is useful? The authors may design a toy example (e.g. the one in [1]) where the analytic gradient and reparametrization gradient are both computable. Such an example can help validate the negative conclusions in the paper.

- A minor comment for the paper's clarity: the review of existing work ends after page 6, which contains weakly relevant information. For example, if the paper learns \beta by Eq 12, it might be distractive to review all three methods to learn \beta, at least before introducing proposed method.

Additional questions:

- The C matrix in Sec. 4.2 is very high dimensional (dim(\lambda) x Jdim(\lambda)) with dim(\lambda) as number of neural net parameters. How scalable is solving Eq 12?
- The paper mentions, "A good example of such a variational distribution q is normalizing flow, where z is the result of pushing forward a base distribution q through a neural network parameterised by λ". The normalizing flow requires the transformation to be invertible, but a neural network is generally non-invertible. Maybe it refers to the semi-implicit distribution [2,3]?

[1] REBAR: Low-variance, unbiased gradient estimates for discrete latent variable models, Tucker et al. 2017

[2] Semi-Implicit Variational Inference, Yin & Zhou, 2018

[3] Unbiased Implicit Variational Inference, Titsias & Ruiz, 2018

---

> ### Author Response · Authors · 2023-10-17
> **Addressing comments from Reviewer CLe6**
>
> ### **The proposed ZVCV-GD does not achieve the optimal objective value in any case. It raises the question of its effectiveness.**
>
> We agree that ZVCV-GD does not effectively improve convergence. This observation aligns with the revised stance of this paper that a control-variate-adjusted gradient estimator is less effective than the simple method of increasing the number of gradient samples.
>
> ### **Since the practitioners often take only a single-sample estimate for the reparameterization gradient, it might hint at the negative results concluded by the paper.**
>
> We agree that taking a single-sample estimate of the reparametrization gradient (RG) is common practice. Our empirical results, however, show that taking multi-sample estimates of the RG improves convergence time. This is a “positive” result. The negative result is that control-variate-adjusted RGs seem ineffective compared to the multi-sample RG.
>
> ### **QuadCV does achieve the best performance and has the lowest ratio. This seems to suggest the variance reduction is useful?**
> It’s true that in Figure 1, QuadCV sometimes achieves the best performance. This is supported by the corresponding plots in Figure 2 which shows QuadCV achieves the lowest ratio. But it is not easy to conclude that the variance reduction was “useful”. Note that the ELBO gain achieved by using QuadCV is extremely small compared to NoCV. For instance, in Fig 1(a), top-right-corner plot, QuadCV achieves around -150 ELBO at convergence while NoCV and ZVCV-GD achieve -157. We don’t think this marginal gain in ELBO is useful (there’s no guarantee that this will translate to a meaningfully better downstream metrics, see e.g. Yao et al. 2018) nor worthwhile (see Figure 3a and 6a which show that the bulk of improvement is coming from the early stage of optimisation). However, we agree that variance reduction is useful in the beginning of the optimisation to improve convergence time and guard against poor initialisation.
>
> Yao, Y., Vehtari, A., Simpson, D., & Gelman, A. (2018, July). Yes, but did it work?: Evaluating variational inference. In *International Conference on Machine Learning* (pp. 5581-5590). PMLR.
>
> ### **It might be distractive to review all three methods to learn $\beta$, at least before introducing proposed method.**
> Given that we have repositioned this paper as a review of control-variate-adjusted pathwise gradient estimators in variational inference, it seems like a good idea to review different options of learning $\beta$, before Section 5 which details the various control variate options $C$. The $\beta$ and $C$ should be seen as two seperate components that can be mixed and matched.
>
> ### **The C matrix in Sec. 4.2 is very high dimensional. How scalable is solving Eq 12?**
> In our implementation of ZVCV-GD, Eq 12 is solved simultaneously when minimising Eq 13 with gradient descent. Note that $C$ is a block-diagonal matrix and the least squares problem in Eq 13 can be broken down into $dim_\lambda$-number of smaller least squares problems, each with a design matrix of size $L \times (J + 1)$ where $L$ and $J$ are the number of gradient samples and control variates respectively. Therefore, the computation cost will scale linearly to the number of neural network parameters $\lambda$. Note that these smaller least squares problems can also be solved independently in parallel.
>
> ### **$z$ is the result of pushing forward a base distribution through a neural network, but neural networks are generally non-invertible.**
> Thank you for giving us the opportunity to clarify this. We should have described a normalizing flow as the result of pushing forward a base distribution through an invertible transformation that is parameterised by neural networks. We did not have in mind semi-implicit distributions when we made this statement.
>
> ### **Reflect the empirical advantages of the proposed method in the simulations.**
> Given the repositioning of the paper as a review on control-variate-adjusted pathwise gradient in variational inference, it is no longer imperative to show the empirical advantage of the proposed ZVCV. Rather, we detail both the strength and weakness of all methods considered. Our experiments lead us to conclude that taking more gradient samples is preferable for a) its simplicity, b) computational efficiency, c) quick convergence albeit to slightly suboptimal ELBO, and d) robustness to poor initialisations.
>
> ### **Improve the writing clarity and accuracy.**
> Thank you for bringing the confusion in our writing to our attention. We have made the necessary corrections to address these.

---

### Author Response · Authors · 2023-10-17
**Summary of changes and additions**

We would like to extend our gratitude to the reviewers for their time and effort in providing feedback on our work. We have taken the reviewers' comments into careful consideration and revised our paper.

All notable changes and additions are highlighted in red for the readers' convenience. Some notable changes to our paper are:

1. We have repositioned the paper to be a review of control-variate-adjusted pathwise gradient estimators for variational inference. The introduction of ZVCV in this context can be viewed as an exercise in completeness. Namely, the existing control variates for pathwise gradient estimators cannot handle the setting where the variational distribution does not have closed-form mean and covariance. Our general aim in this work is to determine if control variates can:
    - Enhance robustness against poor initialisation
    - Significantly improve convergence in terms of wall-clock time

2. We have clarified that variance reduction is most valuable at the beginning of VI optimisation. Our findings suggest that the simplest and most effective way to achieve this is by increasing the number of gradient samples. ZVCV-GD does not seem to reduce variance enough to have a significant impact on convergence time. QuadCV only starts to reduce variance substantially when the VI optimisation is close to convergence — reducing variance at this stage is wasteful as a tiny improvement in ELBO comes with an unjustifiably large increase in computation time. Furthermore, those few nats of ELBO improvement do not guarantee better downstream metrics.

3. We have rerun BNNs + real NVP with a lower learning rate to ensure more stability during training, per the suggestion of reviewer wCDN.

4. Metrics for each individual run (for selected experiments) are now plotted separately to better illustrate the impact of initialisation on VI convergence.

5. We have introduced new experiments that investigate ZVCV-GD with various hyperparameter settings, including second-order ZVCV and different descent steps in the inner gradient descent loop of ZVCV-GD.

6. We have rectified typographical errors and inconsistencies in naming schemes and the bibliography.

---

### Decision · Action_Editor_dT8c · 2023-11-05

**Recommendation:** Reject

**Comment:**

Unfortunately, all three reviewers unanimously voted for rejection of the paper, for the aforementioned reasons.

Reviewer CLe6 argued that the findings are not sufficiently significant: "The negative results are interesting, but they are relatively predictable given the control variates construction and the general use of reparametrization gradient." Reviewer wCDN had concerns about the experimental environment being inconsistent (e.g., by conducting experiments with both unrealistic and reasonable initialization).

The positioning of the paper isn't still clear to any of the reviewers, who found it ambiguous. In its current form, the paper is closer to a methodology paper proposing ZVCV-GD, while the authors claim it is a review paper on control-variate-adjusted pathwise gradient. For a review paper, the paper would be missing important connections with existing approaches and also with existing review papers: (1) there is no discussion of the motivation behind focusing on pathwise gradient variance reduction [Reviewer wCDN]; (2) it remains unclear how it differs from existing review papers on Monte Carlo gradient estimation [Reviewer CLe6]; (3) the paper does not discuss the relevance to other approaches (e.g., better Monte Carlo techniques) [Reviewer wCDN]; and (4) should provide insights on why some of these methods work better than others [Reviewer hVtQ].

**Audience:**

The concerns mentioned above, combined with the poor quality of the presentation (which deteriorated further after the revision), made the paper less interesting for the reviewers, and this might extrapolate to the general TMLR audience. Quoting Reviewer wCDN, this paper "fails to pique the reader's interest".

**Claims And Evidence:**

During the discussion period, the three reviewers were unfortunately unconvinced about this submission. First of all, it remains unclear whether this is a review or a method paper.
+ If positioned as a review paper, which seems to be the authors' intent, it is missing important connections with other methods [Reviewers wCDN and CLe6] and a "deeper understanding of the problem" [Reviewer hVtQ].
+ If positioned as a method paper, the experiments involve inconsistent experimental environments (e.g., initialization) [Reviewer wCDN], and the results are "relatively predictable" [Reviewer CLe6].